nature
structural &
molecular biology
## OPEN

# Higher-order phosphatase–substrate contacts terminate the integrated stress response

Yahui Yan [iD], Heather P. Harding [iD] and David Ron [iD] ✉

Many regulatory PPP1R subunits join few catalytic PP1c subunits to mediate phosphoserine and phosphothreonine dephosphorylation in metazoans. Regulatory subunits engage the surface of PP1c, locally affecting flexible access of the phosphopeptide to the active site. However, catalytic efficiency of holophosphatases towards their phosphoprotein substrates remains unexplained. Here we present a cryo-EM structure of the tripartite PP1c–PPP1R15A–G-actin holophosphatase that terminates signaling in the mammalian integrated stress response (ISR) in the pre-dephosphorylation complex with its substrate, translation initiation factor 2α (eIF2α). G-actin, whose essential role in eIF2α dephosphorylation is supported crystallographically, biochemically and genetically, aligns the catalytic and regulatory subunits, creating a composite surface that engages the N-terminal domain of eIF2α to position the distant phosphoserine-51 at the active site. Substrate residues that mediate affinity for the holophosphatase also make critical contacts with eIF2α kinases. Thus, a convergent process of higher-order substrate recognition specifies functionally antagonistic phosphorylation and dephosphorylation in the ISR.

Signaling in many pathways is terminated by dephosphorylation of pSer and pThr. This task is relegated to complexes between a small number of related protein phosphatase catalytic subunits and hundreds of different regulatory subunits that impart substrate specificity on the holophosphatases[1]. A diverse group of regulatory subunits bind members of the protein phosphatase 1 (PP1) family of related catalytic subunits[2,3].

The structural basis of complex formation between the PPP1R, regulatory, and PP1c, catalytic, subunits is understood. The former engage the surface of PP1c via short linear segments, exemplified by the conserved RVxF, φφ and 'R' motifs observed in complexes of $G_M$[4], spinophilin[5], PNUTS[6] or Phactr1 (ref. [7]) regulatory subunits with PP1c. Regulatory subunit binding sculpts the surface of PP1c to bias substrate access to the active site. This explains local selectivity for and against flexible pSer/pThr-bearing substrate-derived peptides[2,8–10], but cannot fully account for the catalytic efficiency of holophosphatases directed towards their physiological substrates—globular domains of phosphoproteins.

Eukaryotes share a signal transduction pathway that couples changing rates of translation initiation to a pervasive transcriptional and translational program. Known in yeasts as the general control response[11] and in animals as the integrated stress response (ISR)[12], the ISR is triggered by phosphorylation of Ser51 of the α-subunit of translation initiation factor 2 (eIF2α[P]), an event mediated by four known kinases (GCN2, PERK, HRI and PKR). ISR manipulation reveals its broad role in cellular homeostasis and the potential for tuning the response to therapeutic ends[13].

Dephosphorylation of eIF2α[P] terminates signaling in the ISR. In mammals, dephosphorylation is assisted by one of two known regulatory subunits, PPP1R15A or PPP1R15B. PPP1R15A, also known as GADD34, is encoded by an ISR-inducible gene and serves in negative feedback[14–17], whereas PPP1R15B, also known as CReP, is constitutively expressed[18]. Genetic studies demonstrate the benefits of extending the ISR and suggest the therapeutic potential of inhibiting eIF2α dephosphorylation[19,20]. Attaining this goal requires a detailed understanding of the enzyme(s) involved.

The mammalian PPP1R15s are proteins of more than 600 amino acids, but their conserved portion is limited to ~70 residues near their C termini. This segment (residues 555–621 in human PPP1R15A), which is sufficient to promote eIF2α[P] dephosphorylation when expressed in cells[15,18], is also conserved in the single PPP1R15 gene of other animal phyla[21] and in a pathogenicity gene of herpesviruses[22]. The N-terminal half of this conserved PPP1R15 core binds PP1c[15,23–25]. However, the PP1c–PPP1R15 complex is little better at dephosphorylating eIF2α[P] than a PP1c apoenzyme[25,26]. High-resolution crystal structures of PP1c in complex with PPP1R15A[24] and PPP1R15B[25] exist, however, these have merely generic features of PP1c holoenzymes. Addition of G-actin strongly stimulates eIF2α[P] dephosphorylation by PP1c–PPP1R15. G-actin binds the C-terminal half of the conserved core of the PPP1R15s to form a ternary PP1c–PPP1R15–G-actin complex both in vitro[25,27] and in cell extracts[28], but the basis of its stimulatory activity remains unknown.

Here we report on a crystal structure of the binary PPP1R15A–G-actin complex and on a cryo-EM structure of the tripartite PP1c–PPP1R15A–G-actin holophosphatase in complex with its substrate. Comparison of this eIF2α[P] pre-dephosphorylation complex with a counterpart eIF2α pre-phosphorylation complex (with kinase PKR[29]) reveals similar principles by which a protein kinase and a PP1-based holophosphatase attain substrate-specific catalytic efficiency.

## Results

**Crystal structure of the binary PPP1R15A–G-actin complex.** Enzymatic and binding experiments suggested that the conserved core of PPP1R15s bind PP1c and G-actin via nonoverlapping linear segments (Fig. 1a). High-resolution crystal structures of binary PPP1R15A–PP1 and PPP1R15B–PP1 complexes exist[24,25], but similar information on contacts with G-actin is missing. We exploited the observation that G-actin can be recruited to the PPP1R15-containing holophosphatase in complex with DNase I[25]. DNase I stabilizes actin monomers[30] and facilitates crystallization[31]. DNase I–G-actin formed a stable complex with the C-terminal

---

Cambridge Institute for Medical Research, University of Cambridge, Cambridge, UK. ✉e-mail: dr360@medschl.cam.ac.uk

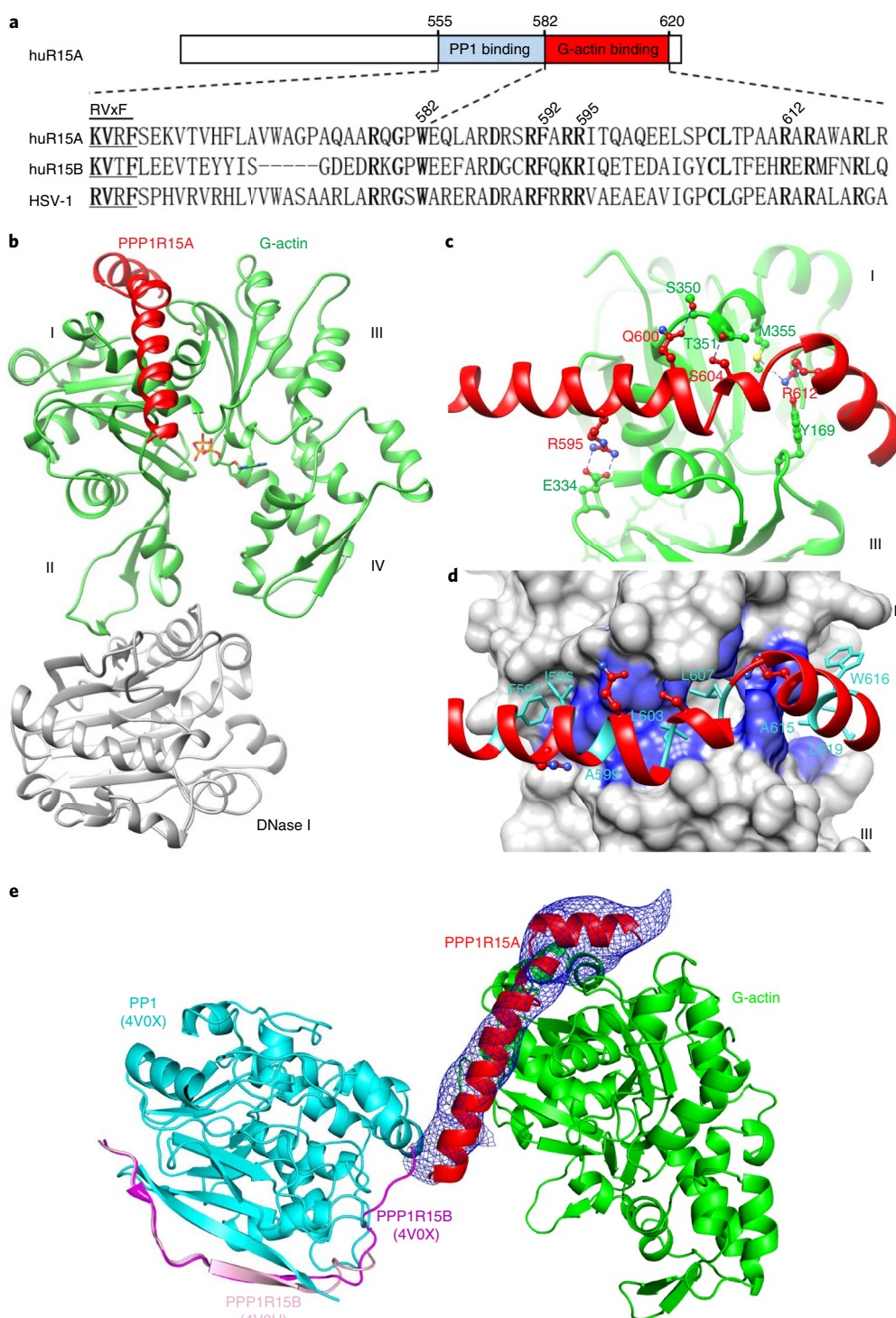

**Fig. 1 | PPP1R15A engages the barbed end of G-actin. a**, Schema of human PPP1R15A. The PP1- and G-actin-binding segments are marked. Beneath is an alignment of human PPP1R15A (huR15A), human PPP1R15B (huR15B) and the regulatory subunit of herpes simplex virus (HSV-1). Conserved residues are in bold. The 'RVxF' motif, common to many PPP1Rs, is noted. Numbering refers to human PPP1R15A. **b**, Ribbon diagram of the overall crystallographic structure of the PPP1R15A–G-actin–DNase I complex at a resolution of 2.55 Å. Actin domains are numbered (I–IV). **c**, Close-up view of hydrogen-bonding interactions between PPP1R15A (red) and G-actin (green). The density map for PPPR15A is shown in Extended Data Fig. 1. **d**, Hydrophobic interactions of G-actin (blue surface) with the indicated PPP1R15A residues (side chains as turquoise sticks). **e**, Model of a tripartite holophosphatase constructed by aligning the PPP1R15A–G-actin–DNase I complex (above) and the binary PP1G–PPP1R15B complex (PDB 4V0X) to the low-resolution PP1G–PPP1R15B–G-actin complex (PDB 4V0U); the former via G-actin and the latter via PP1. Shown is the PP1G–PPP1R15B from PDB 4V0X (with a root mean squared deviation (r.m.s.d.) of 0.318 Å between the 290 Cα pairs of PP1c) and the PPP1R15A–G-actin from the PPP1R15A–G-actin–DNase I complex (with r.m.s.d. of 0.626 Å between 343 Cα pairs of actin). DNase I is omitted for clarity. Note the proximity of the C terminus of PPP1R15B and the N terminus of PPP1R15A (consisting of the same residue of both orthologs: PPP1R15B Trp 662 and PPP1R15A Trp 582) in the two binary complexes. The previously unaccounted density in the barbed end of G-actin (displayed as an average difference electron density from PDB 4V0U and shown as a blue mesh) accommodates the actin-binding helices of PPP1R15A (red) from the aligned PPP1R15A–G-actin–DNase I complex.

### Table 1 | X-ray data collection and refinement statistics

| | G-actin–DNase I–PPP1R15A[582-621] complex (PDB 7NXV) |
|---|---|
| **Data collection** | |
| Space group | |
| Cell dimensions | |
| $a, b, c$ (Å) | 86.48, 107.79, 192.42 |
| $\alpha, \beta, \gamma$ (°) | 90.00, 90.00, 90.00 |
| Resolution (Å) | 55.12–2.55 (2.62–2.55)* |
| $R_{merge}$ | 0.315 (1.701) |
| $I/\sigma I$ | 6.5 (1.4) |
| Completeness (%) | 100.0 (100.0) |
| Redundancy | 6.9 (7.1) |
| **Refinement** | |
| Resolution (Å) | 2.55–55.11 |
| No. reflections | 56,450 |
| $R_{work}/R_{free}$ | 0.216 / 0.248 |
| No. atoms | |
| Protein | 10,476 |
| Ligand/ion | 118/6 |
| Water | 50 |
| B factors | |
| Protein | 39.15 |
| Ligand/ion | 41.59/40.71 |
| Water | 23.72 |
| R.m.s. deviations | |
| Bond lengths (Å) | 0.002 |
| Bond angles (°) | 1.209 |

*Values in parentheses are for highest-resolution shell. The dataset was collected from a single crystal.

portion of PPP1R15A, which crystallized and diffracted to a resolution of 2.55 Å (Fig. 1b–d, Extended Data Fig. 1a and Table 1).

Residues 582–619 of PPP1R15A form two helices separated by a sharp turn that wraps around G-actin, engaging the groove between subdomains I and III. DNase I binds the opposite end of G-actin, whereas D-loop insertion in F-actin and G-actin binding by small molecules, such as cytochalasin D, overlaps the PPP1R15A site (Extended Data Fig. 1b,c). These findings explain the ability of cytochalasin to antagonize the stimulatory effect of G-actin on eIF2α[p] dephosphorylation in vitro[25], and the inhibitory effect of drug-enforced actin polymerization on dephosphorylation in cells[28].

The location of PPP1R15A on the surface of G-actin corresponds to a site of electron density observed in a crystal structure of the PP1G–PPP1R15B–G-actin complex (PDB 4V0U)[25]. Due to the low resolution of the map (7.8 Å), the density remained unassigned. However, aligning G-actin in the ternary PP1G–PPP1R15B–G-actin (PDB 4V0U) and binary PPP1R15A–G-actin–DNase I complex (as here) reveals that the PPP1R15A from the high-resolution binary complex fits nicely in the unassigned density in the groove of G-actin in PDB 4V0U. Furthermore, the C-terminal PPP1R15B Trp662 from a high-resolution binary PP1G–PPP1R15B complex (PDB 4V0X, aligned by its PP1 to PDB 4V0U) is close enough to the N-terminal PPP1R15A orthologous residue (Trp582) in its complex with G-actin (as here) to complete a PPP1R15 peptide chain in the ternary complex and provide a view of a composite PPP1R15B/A-containing tripartite holoenzyme (Fig. 1e).

**Validation of the role of actin in eIF2α[p] dephosphorylation.** Residues lining one face of the PPP1R15A helical extension form hydrophobic interactions and hydrogen bonds with G-actin (Fig. 1c,d). Human PPP1R15A Phe592 inserts into a hydrophobic cavity on the surface of actin, whereas Arg595 forms a salt bridge with actin Asp334 and PPP1R15A Arg612 forms hydrogen bonds with actin Tyr169 and Met355. These PPP1R15A residues are conserved across species and were selected for functional studies by mutagenesis.

Recombinant wild type (WT) or mutant human PPP1R15A core fragment (residues 553–624) in complex with PP1A and G-actin were compared in dephosphorylating the N-terminal lobe of eIF2α[p]. A kinetic defect was observed in all four mutants tested (Fig. 2a,b). The defect was selective for actin-containing mutant holophosphatases—the low baseline activity of the apoenzymes (lacking G-actin) was barely affected by the mutations. The strongest defect was observed in double mutants compromising both PPP1R15A Phe592 and Arg595; their catalytic efficiency ($k_{cat}/K_m$) was nearly 50-fold less than the wild type, approaching that of the apoenzyme (lacking G-actin, Fig. 2c). The more C-terminal contact with G-actin, mediated by Arg612, proved less important.

**Fig. 2 | Actin-facing mutations disrupt PPP1R15A action in vitro and in cells. a**, Coomassie-stained PhosTag gels resolving phosphorylated from nonphosphorylated eIF2α following eIF2α[p] dephosphorylation in vitro with WT or mutant PPP1R15A holophosphatases. The concentration of binary PPP1R15A–PP1A and G-actin is indicated, as is the genotype of the PPP1R15A component. Shown is a representative example of experiments reproduced at least three times. **b**, Time-dependent progression of dephosphorylation reactions described in **a** (showing all replicates). The dotted line is fit to a first-order decay. **c**, Best fit of three independent experiments in panel **b** ±95% confidence intervals of the $k_{cat}/K_m$ of the indicated PPP1R15A–PP1A pairs in reactions lacking (apo) and containing (holo) G-actin. **d**, Plot of the association-phase plateau BLI signal arising from a probe consisting of immobilized WT or mutant PPP1R15A derivatives interacting with the indicated concentration of G-actin in solution. **e**, Two-dimensional flow cytometry contour plots (with outliers) of untreated and thapsigargin (Tg)-treated CHOP::GFP transgenic CHO-K1 cells transfected with mCherry alone or full-length WT or mutant mouse PPP1R15A fused to mCherry at their C termini. The gate of the mCherry⁺ cells analyzed in panels **f** and **g** is indicated in gray. Note the lack of effect of mCherry on the CHOP::GFP ISR marker, the attenuation of CHOP::GFP induction in cells transfected with wild-type PPP1R15A::mCherry and defective attenuation by mutations in G-actin-facing residues in PPP1R15A. **f**, Box (25th and 75th percentile) and whiskers plots (cut-off at the 1–99th percentiles) of the CHOP::GFP signal in the mCherry⁺ experimental populations indicated in gray (cell numbers shown in **e**). **g**, Individual data points, mean ± s.d. of fold ISR attenuation by the indicated PPP1R15A::mCherry fusions from replicate experiments as in **e**. (P values for two-tailed t-test comparisons shown; n = 4 independent experiments for F585A and F585A;R588A mutants and n = 5 for the other samples. NS, not significant.) **h**, Fluorescent photomicrographs of CHO-K1 cells transfected with expression plasmids encoding the stress granule marker G3BP-GFP and mCherry alone, WT or mutant mouse PPP1R15A[F585A;R588E]::mCherry fusion proteins (as in **e**). Cells were treated with 0.5 mM sodium arsenite to induce stress granules (white arrows). Shown are representative images reproduced three times. **i**, Percentage of mCherry⁺GFP⁺ (C⁺G⁺) transfected cells with stress granules (SG) in each high-power field in sodium arsenite-treated cells as in '**h**'. All data points, mean ± s.d. (P values for two-tailed t-test comparisons shown, n = 9 biological replicates for mCherry, 10 for WT and 18 for F585A;R588E). Results are representative of experiments reproduced three times. Source data for **a–d**, **f**, **g** and **i** are available online.

The lack of response of the mutants to G-actin was confirmed in titration: whilst binary complexes of PP1A and wild-type PPP1R15A responded to G-actin with a vigorous increase in activity, PPP1R15A[R595E] alone, and even more so in conjunction with a PPP1R15A[F592A] mutation, markedly attenuated the responsiveness of the binary complex to G-actin (Extended Data Fig. 2). The attenuated response of the mutant binary complexes in the enzymatic assay correlated with defective G-actin binding, as assessed by biolayer interferometry (BLI) (Fig. 2d).

This new information on functionally important contacts between PPP1R15A and G-actin motivated us to revisit the role of actin in eIF2α[p] dephosphorylation in cells. PPP1R15

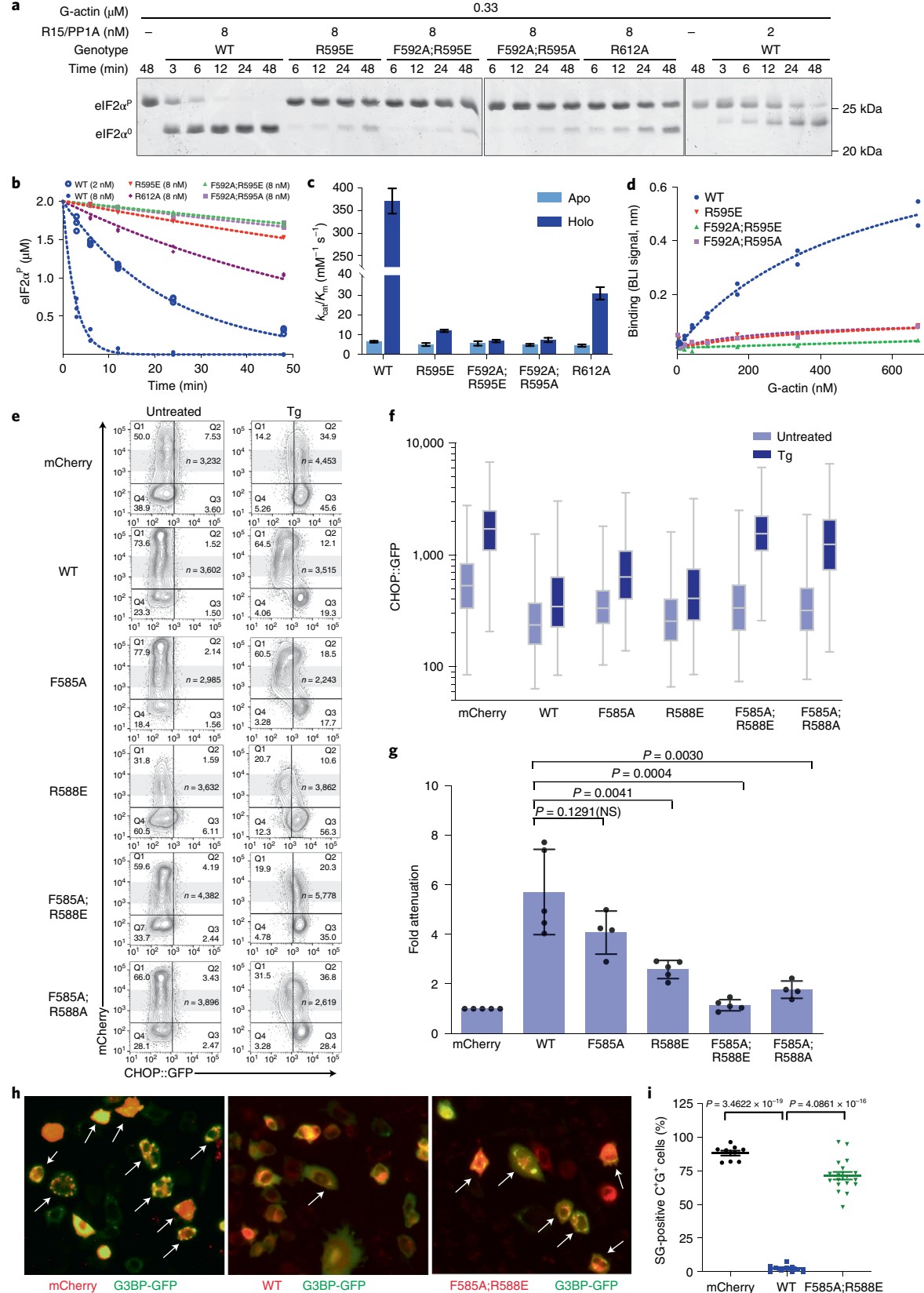

expression-mediated eIF2α[P] dephosphorylation attenuates the ISR[15,18]. This feature was exploited to examine the effect of the actin-facing PPP1R15A mutations in cells. We used a cell-based assay in which ISR activity was monitored by a CHOP::GFP fluorescent reporter[15]. Treatment with thapsigargin, an agent that activates the ISR by triggering PERK-dependent eIF2α phosphorylation, activated the CHOP::GFP reporter, as reflected by a shift to the right in the distribution of the fluorescent signal detected by flow cytometry. Coexpression of a control plasmid encoding mCherry alone had no effect on the CHOP::GFP signal. Expression of wild-type full-length mouse PPP1R15A (fused to mCherry at its C terminus) attenuated the thapsigargin-mediated CHOP::GFP signal[25]. Mutant mouse PPP1R15A[R588E] and PPP1R15A[F585A] (counterparts to human PPP1R15A[R595E] and PPP1R15A[F592A]) had a weaker attenuating effect on the ISR marker, a defect that was most conspicuous in the mouse PPP1R15A[F585A;R588E] double mutant (Fig. 2e–g). The mouse PPP1R15A[R605A] mutation had no observable effect in this assay, consistent with the weaker effect of the counterpart human PPP1R15A[R612A] mutation in vitro (Fig. 2a).

Stress granule formation is a convenient orthogonal marker of ISR activation[32]. Consistent with this notion, the abundance of stress granules in stressed cells was attenuated by enforced expression of wild-type full-length mouse PPP1R15A, but less so by expression of PPP1R15A with G-actin-facing mutations (Fig. 2h,i).

The effects of the actin-facing mutations are consistent with G-actin contributing to the in vivo activity of a holophosphatase consisting of full-length PPP1R15A. To explore this further we compared β-actin that was wild type at its PPP1R15A-contacting surface with a charge-reversal mutant β-actin[E334R] (a residue contacting human PPP1R15A Arg595, Fig. 1c) for their ability to restore ISR inhibition when coexpressed with the severe, charge-reversed full-length mutant mouse PPP1R15A[F585A;R588E]. To favor retention of the exogenous β-actin in its G-form, a polymerization-deficient variant was used (β-actin[A204E;P243K])[33]. Neither enforced expression of otherwise wild type, nor mutant β-actin[E334R] affected the ability of wild-type full-length PPP1R15A to attenuate the ISR. However, β-actin[E334R] consistently (albeit partially) reversed the defect in mouse PPP1R15A[F585A;R588E]-mediated suppression of the ISR (red arrow, Fig. 3a–c).

**Fig. 3 | Allele-specific suppression of a PPP1R15A surface-charge mutation by a reciprocal surface-charge mutation in β-actin. a**, Two-dimensional flow cytometry dot plots of thapsigargin-treated cells transfected with plasmids encoding either blue fluorescent protein (BFP) alone, BFP linked (in-*trans*) to an otherwise wild-type (polymerization-deficient A204E;P243K) β-actin or BFP linked to polymerization-deficient β-actin[E334R] surface-charge reversal mutant (affecting a residue that forms a salt bridge with mouse PPP1R15A[R588]). Cells were cotransfected with plasmids encoding mCherry alone or the indicated mouse PPP1R15A::mCherry fusion proteins (as in Fig. 2e–g). Shown are the mCherry and GFP channels of the β-actin-expressing (BFP+) populations in each dataset. β-Actin[E334R] selectively restores the ability of the PPP1R15A[F585A;R588E] mutation to attenuate the ISR (red arrow), whereas both the wild type and β-actin[E334R] enhance ISR attenuation by PPP1R15A[F585A;R588A] (green arrows). **b**, Box (25th and 75th percentile) and whiskers plots (cut-off at the 1–99th percentiles) of the CHOP::GFP signal in untreated (−) and thapsigargin-treated (+) from the BFP+, mCherry+ cells marked by gray shading in **a** above (cell numbers shown in **a**). The signal in the critical sample coexpressing PPP1R15A[F585A;R588E] and β-actin[E334R] is denoted by a horizontal red arrow. **c**, Individual data points and mean ± s.d. of the ISR attenuation factor of the indicated subset of PPP1R15A::mCherry fusions from replicate experiments as in **a** (P values for two-tailed *t*-test comparisons shown, n = 4 independent experiments for mCherry, WT and F585A;R588E and n = 3 for W575A and F585A;R588A). Source data for **a–c** are available online.

Genetic rescue by the charge-substituted β-actin[E334R] was PPP1R15A allele-specific: both wild type and β-actin[E334R] modestly but similarly enhanced the activity of the mouse PPP1R15A[F585A;R588A] mutant (green arrows, Fig. 3a), whereas neither reversed the defect in the mouse PPP1R15A[W575A] mutant (the counterpart to human PPP1R15A[W582]). These differences in response of the PPP1R15A mutants to actin are consistent with charge complementarity of the reciprocal β-actin[E334R]/PPP1R15A[R588E] pairing, and charge exclusion of the wild-type β-actin Asp334/PPP1R15A[R588E] pairing accounted for the selective (albeit partial) genetic rescue in the case of the

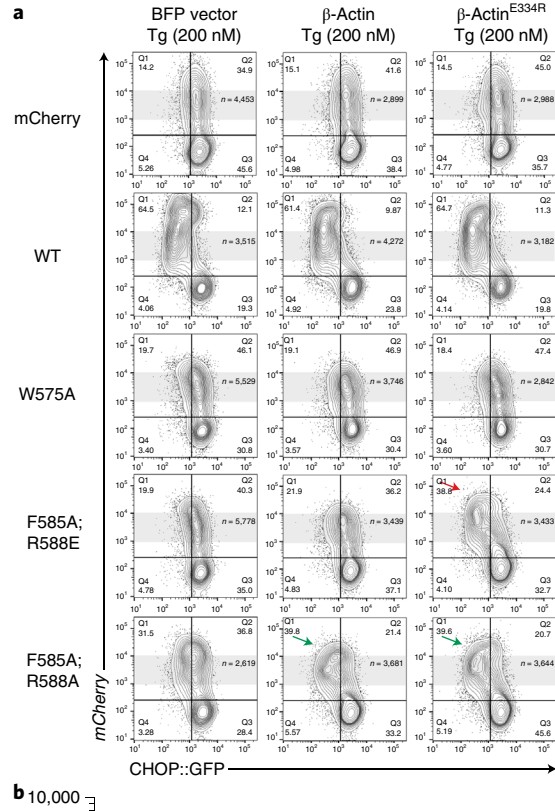

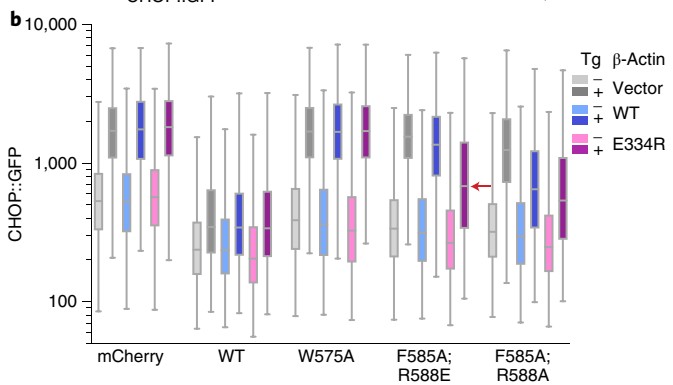

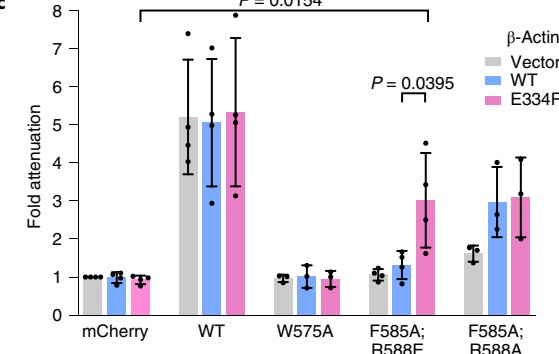

**Table 2 | Cryo-EM data collection, refinement and validation statistics**

| | PP1A$^{D64A}$–PPP1R15A$^{553-624}$–G-actin–DNase I–eIF2α$^P$-NTD (EMD-12665) (PDB 7NZM) |
|---|---|
| **Data collection and processing** | |
| Magnification | ×130,000 |
| Voltage (kV) | 300 |
| Electron exposure (e⁻/Å²) | 46.84 |
| Defocus range (μm) | −2.8 to −1.0 |
| Pixel size (Å) | 0.652 (counted super-resolution 0.326) |
| Symmetry imposed | C1 |
| Initial particle images (no.) | 132,495 |
| Final particle images (no.) | 60,413 |
| Map resolution (Å) | 3.96 |
| FSC threshold | 0.143 |
| Map resolution range (Å) | 3.5 to 6.4 |
| **Refinement** | |
| Initial model used (PDB code) | 4MOV, 2A42 and 1KL9 |
| Model resolution (Å) | 4.1 |
| FSC threshold | 0.5 |
| Model composition | |
| Nonhydrogen atoms | 9,109 |
| Protein residues | 1,157 |
| Ligands | 4 |
| *B* factors (Å²) | |
| Protein | 47.19 |
| Ligand | 54.66 |
| R.m.s. deviations | |
| Bond lengths (Å) | 0.006 |
| Bond angleesolves (°) | 1.078 |
| Validation | |
| MolProbity score | 1.81 |
| Clashscore | 10.99 |
| Poor rotamers (%) | 0 |
| Ramachandran plot | |
| Favored (%) | 96.24 |
| Allowed (%) | 3.76 |
| Disallowed (%) | 0 |

former. The mouse PPP1R15A$^{R588A}$ mutation was indifferent to the surface charge of the exogenous β-actin, accounting for the similar ability of increased concentrations of wild type or β-actin$^{E334R}$ to partially restore its function by mass action. The mouse PPP1R15A$^{W575A}$ mutation, which affects PPP1R15A by a different mechanism (see below) and binds G-actin with wild-type affinity[27], was indifferent to an increase in concentration of β-actin, wild type or mutant.

**Structure of the pre-dephosphorylation complex.** To better understand the catalytically efficient tripartite holophosphatase, we sought to trap a pre-dephosphorylation complex for structural analysis. To this end, we exploited observations that a PP1A$^{D64A}$ mutation in the active site attenuates dephosphorylation and facilitates trapping of enzyme–substrate pre-dephosphorylation complexes[34].

The substrate (the N-terminal lobe of eIF2α$^P$) coeluted with a catalytically deficient PP1A$^{D64A}$–PPP1R15A–G-actin tripartite holophosphatase in size-exclusion chromatography, and the complex gave rise to well-defined particles in cryo-EM (Extended Data Fig. 3a,b and Table 2).

A cryo-EM map at an overall resolution of 3.96 Å was reconstructed and one copy of each component, PP1A$^{D64A}$–PPP1R15A–G-actin–DNase I–eIF2α$^P$, was well resolved in the map (Fig. 4a). PP1A is bound by the conserved N-terminal segment of PPP1R15A (residues 560–582 are resolved). High-resolution crystal structures of PP1A–PPP1R15A (PDB 4XPN) and PP1G–PPP1R15B (PDB 4V0X) readily dock in the cryo-EM map. The PPP1R15A chain, truncated in the crystal structure at Ala568, can be completed in the cryo-EM map and is seen to follow the known trajectory of PPP1R1B on the surface of PP1 (Extended Data Fig. 3d). The crystal structure of the binary PPP1R15A–G-actin (with the bound DNase I) also docks comfortably in the cryo-EM density, and the continuity of a composite PPP1R15B>A chain is readily established in the consecutively docked crystallographic structures (Extended Data Fig. 3d). The β-barrel of eIF2α engages one face of the kinked helical extension of PPP1R15A (the other face of which engages G-actin) (Fig. 4b) and the substrate loop containing pSer51 is inserted into the active site of the mutant PP1A$^{D64A}$ (Fig. 4c and Extended Data Fig. 3e).

Although present in the crystallized constructs, residues that are C-terminal to PPP1R15B Trp662 (the counterpart of PPP1R15A Trp582) were invariably disordered in binary complexes with PP1c[24,25]. This suggests an important role for G-actin binding in stabilizing the helical conformation of this segment. Engagement of the β-barrel of eIF2α by this helical assembly and the attendant positioning of the substrate loop in the active site of the holoenzyme, provide a structural explanation for the role of G-actin as a cofactor in eIF2α$^P$ dephosphorylation.

Alignment of PP1G–PPP1R15B (PDB 4V0X) to the PP1c of the pre-dephosphorylation complex reveals a conspicuous difference in the location of an invariant PPP1R15 Trp residue: in the binary complex, PPP1R15B Trp662 engages a pocket on the surface of PP1 (also bound by Phactr1 Trp542 in the PP1–Phactr1 complex[7], PDB 6ZEE). In the pre-dephosphorylation complex the corresponding residue, PPP1R15A Trp582, has flipped into a different position on the surface of PP1c (Fig. 4d). Enforcing the 'Trp up' position, observed in the binary PP1G–PPP1R15B complex on PPP1R15A in the holoenzyme, would swing the helical assembly upwards to clash with the substrate (Fig. 4d). The 'Trp down' position, observed in the pre-dephosphorylation complex, is stabilized locally by hydrogen bonds between the Nε atom of Trp582, the PP1A Glu139 side chain and the carbonyl oxygens of Gly97 and Lys98, bringing the helical assembly into a conformation compatible with substrate recruitment and catalysis. These findings provide a structural explanation for observation that, although it makes no measurable contribution to PP1's or G-actin's binding kinetics to PPP1R15 (ref. [27]), this invariant Trp residue has an important role in substrate dephosphorylation (see ref. [25] and below).

Plasticity of the holoenzyme is also supported by comparing the crystal structure of the PP1G–PPP1R15B–G-actin holophosphatase (PDB 4V0U) with the cryo-EM structure of the pre-dephosphorylation complex. When aligned by their PP1c regions, the G-actin-stabilized helical assembly of the crystallized holoenzyme is tilted ~18° towards the substrate (as observed in the pre-dephosphorylation complex), thus clashing with it (Supplementary Video 1). The pivot of the tilt corresponds to the region of PPP1R15B Trp662, although the resolution of the crystal structure is too low to assign a specific 'up' or 'down' position to its side chain. These observations show the potential for the holoenzyme to assume both catalytically competent and incompetent conformations.

eIF2α[P] binding in the competent conformation positions the pSer51 substrate loop in the PP1[D64A] mutant active site. Density consistent with a phosphate is attached to Ser51, marking this as a pre-dephosphorylation complex. The Ser51 phosphoryl group, which occupies the position of a phosphate ion commonly observed in other PP1c structures, interacts with surrounding PP1A residues (His66, Arg96, Asn124, His125, Arg221, His248) (Fig. 4c). As expected, the M1 metal that is coordinated by PP1 Asp64 (and aligns the attacking water molecule in the wild-type enzyme[34,35]) is missing in this PP1A[D64A] mutant complex. Instead, the side chain of PP1A His248 moves from its more distal position, found in bimetallic PP1s, to contact a phosphate oxygen of eIF2α[P] here.

There are no known structures of isolated eIF2α[P]. The substrate loop is disordered in the pre-phosphorylation complex with PKR (PDB 2A19). In complexes of eIF2(α[P]) with its nucleotide exchange factor eIF2B, pSer51 faces the interior of the loop, a conformation that may be stabilized by interactions with surrounding eIF2B residues (PDB 6O9Z[36], 6I3M[37], 6JLZ[38], 7D43[39]). Ser51 is also inward facing in NMR solution structures of isolated nonphosphorylated eIF2α (PDB 1Q8K)[40], suggesting this to be the favored conformation of the substrate loop. Contacts observed here between the loop backbone and PP1 residues Tyr134, Arg221 and Arg96 (Fig. 4c and Extended Data Fig. 3e,f) may play a role in stabilizing its extended, catalytically competent conformation, with pSer51 facing outwards: but in this, there seems to be no direct role for PPP1R15.

The role of PPP1R15 in promoting catalysis is played out by enabling distant enzyme–substrate interactions. These are contingent on G-actin-mediated stabilization of the helical assembly of PPP1R15A. Binding of G-actin to one face of the helix presents its other face to the conserved β-barrel of the substrate, enabling several interactions: PPP1R15A Arg591 is buried between eIF2α Met44 and Tyr81 and forms a salt bridge with Asp83. PPP1R15A Arg594 forms hydrogen bonds with eIF2α Tyr32 and a salt bridge with Asp42. PPP1R15A Arg587 forms a salt bridge with eIF2α Asp83 and eIF2α Lys79 is within range of salt bridges with PPP1R15A Asp588 and G-actin Asp25 (Fig. 4b). These substrate interactions are limited to the N-terminal half of the kinked helical assembly of PPP1R15A, which may explain the greater impact of mutations affecting N-terminal contacts with G-actin (PPP1R15A[F592A] and PPP1R15A[R595E/A]) compared with perturbations affecting C-terminal contacts (PPP1R15A[R612A] or cytochalasin D) (Fig. 2 and ref. [25]). Although absent from the complexes assembled here, the C-terminal lobe of eIF2α and the attached eIF2β and γ subunits are readily accommodated in the structure (Extended Data Fig. 4)

**Validation of substrate recognition by the holophosphatase.** To examine the functional importance of the enzyme–substrate contacts noted above, we focused on three mutations: PPP1R15A[R591A] and PPP1R15A[R594A], predicted on structural grounds to disrupt contacts with the β-barrel of eIF2α, and PPP1R15A[W582A], predicted

to destabilize the active conformation of the holophosphatase. Mutation of any one of these three residues to alanine weakened the holophosphatase in vitro, with PPP1R15A[R591A] and PPP1R15A[W582A] having the strongest effect (Fig. 5a–c). In cells also, the corresponding mutations, mouse PPP1R15A[R584A] and PPP1R15[W575A], attenuated the ability of the expressed mouse PPP1R15A to block the ISR, or interfere with stress granule formation (Fig. 5d–g).

The formation of a pre-dephosphorylation complex gave rise to a robust binding signal in BLI, with biotinylated eIF2α[P] (immobilized on the probe) interacting with a mutant PP1A[D64A]–PPP1R15A–G-actin holophosphatase (in solution) (Fig. 5h). Complex formation depended on G-actin, consistent with the latter's role in stabilizing the substrate-binding conformation of the holophosphatase. When used as a probe, biotinylated nonphosphorylated eIF2α[0] also formed a complex with the holophosphatase, but it was less stable. Importantly, the PPP1R15A[W582A] and PPP1R15A[R591A] mutations, which interfered with the enzyme's catalytic efficiency, also interfered with substrate binding in BLI (Fig. 5i).

Contacts present in the pre-dephosphorylation complex involve eIF2α residues previously observed to interact with the kinase PKR in a prephosphorylation enzyme/substrate complex[29]. Notably, the side chain of PKR Phe489 is inserted into the cleft formed by eIF2α Met44 and Tyr81 (a counterpart to PPP1R15A Arg591, whose mutation blocked dephosphorylation, Fig. 5a). eIF2α Lys79 forms a salt bridge with PKR Glu490 in the prephosphorylation complex and is within range of salt bridge formation with PPP1R15A Asp588 and actin Asp25, in the pre-dephosphorylation complex, whereas eIF2α Asp83 hydrogen bonds with PPP1R15A Arg591 in the pre-dephosphorylation complex (Fig. 6a) and with the backbone nitrogen of PKR Ala488 in the latter's substrate-binding helix G[29].

In keeping with the functional importance of these contacts, we observed that eIF2α[M44A] and eIF2α[Y81A] mutations interfered with the ability to serve as a substrate of both the PPP1R15A-containing holophosphatase and the ISR-inducing kinase PERK, in vitro. The eIF2α[K79A] mutation also affected both reactions: dephosphorylation more than phosphorylation (Fig. 6b,c). The effect of the eIF2a[D83A] mutation on dephosphorylation could not be tested, as it blocked all phosphorylation (Fig. 6c). Together, these observations point to convergence of higher-order enzyme–substrate contacts in opposing reactions affecting eIF2α.

## Discussion

The structure of the PPP1R15A-based pre-dephosphorylation complex presented here rationalizes the role of G-actin as an essential component of the holophosphatase in vitro. Corroborating biochemical studies and allele-specific suppression of a defective PPP1R15A by a rationally designed compensatory mutation in β-actin, also links G-actin to holophosphatase function in vivo. Actin-dependent contacts between the holophosphatase and a distant surface of eIF2α position the pSer51 at the enzyme active site,

**Fig. 4 | A cryo-EM structure of the eIF2α[P] pre-dephosphorylation complex. a**, Overview of the cryo-EM structure of the pre-dephosphorylation complex (rendered as a GSFSC map in cryoSPARC). The threshold was set to keep the enclosed volume continuous in UCSF Chimera. Constituent proteins labeled. eIF2α[P]-NTD refers to its phosphorylated N-terminal domain (NTD). **b**, Close-up view of contacts between the β-barrel of eIF2α[P] and the non-actin-facing surface of PPP1R15A from the cryo-EM structure. Dotted lines mark hydrogen-bonding interactions. The density map for PPP1R15A is shown in Extended Data Fig. 3d. **c**, Close-up view of the eIF2α[P] substrate loop (blue) in the PP1A active site (cyan). Substrate-contacting active site residues are indicated, as are eIF2α[P] pS51 and the mutated PP1A[D64A]. The gray ball denotes the M2 metal ion. The density map for the eIF2α[P] substrate loop (residues 48–53) is shown in Extended Data Fig. 3e. **d**, Surface view of PP1A (colored by charge for orientation) with PPP1R15A (in red), G-actin and eIF2α[P] in ribbon diagram (all from the cryo-EM structure). PPP1R15B from the binary PP1G–PPP1R15B complex (PDB 4V0X, aligned to the cryo-EM structure by PP1G) in yellow and PPP1R15A from the crystal structure of its binary complex with G-actin–DNase I (as here, aligned to the cryo-EM structure by G-actin) in tan. Note the different disposition of PPP1R15B Trp662 (from the binary PP1G–PPP1R15B complex) and its counterpart, PPP1R15A Trp582 (from the cryo-EM structure of the pre-dephosphorylation complex). The 'Trp up' disposition (exemplified by PPP1R15B Trp662, yellow) favors the PPP1R15 helix (C-terminal to it) as found in the binary PPP1R15A–G-actin–DNase I complex (shown in tan), where it would clash with eIF2α[P]. **e**, Superposition of the cryo-EM pre-dephosphorylation complex (colored as in **a**) and the crystal structure of the PP1G–PPP1R15B–G-actin complex (PDB 4V0U, in ivory and gray) aligned by PP1c. Note the ~18° rotation of the G-actin-bound PPP1R15B helical assembly, which would lead it to clash with eIF2α[P].

paralleling distal contacts conserved in the pre-phosphorylation complex. Thus, the kinases that initiate, and the phosphatases that terminate, signaling in the ISR appear to have converged on a common solution for catalytic efficiency, one that relies on recognizing features of their folded globular substrate, distinct from the substrate loop.

The enzyme studied here contains the conserved core common to PPP1R15s from all phyla. Likewise, the independently

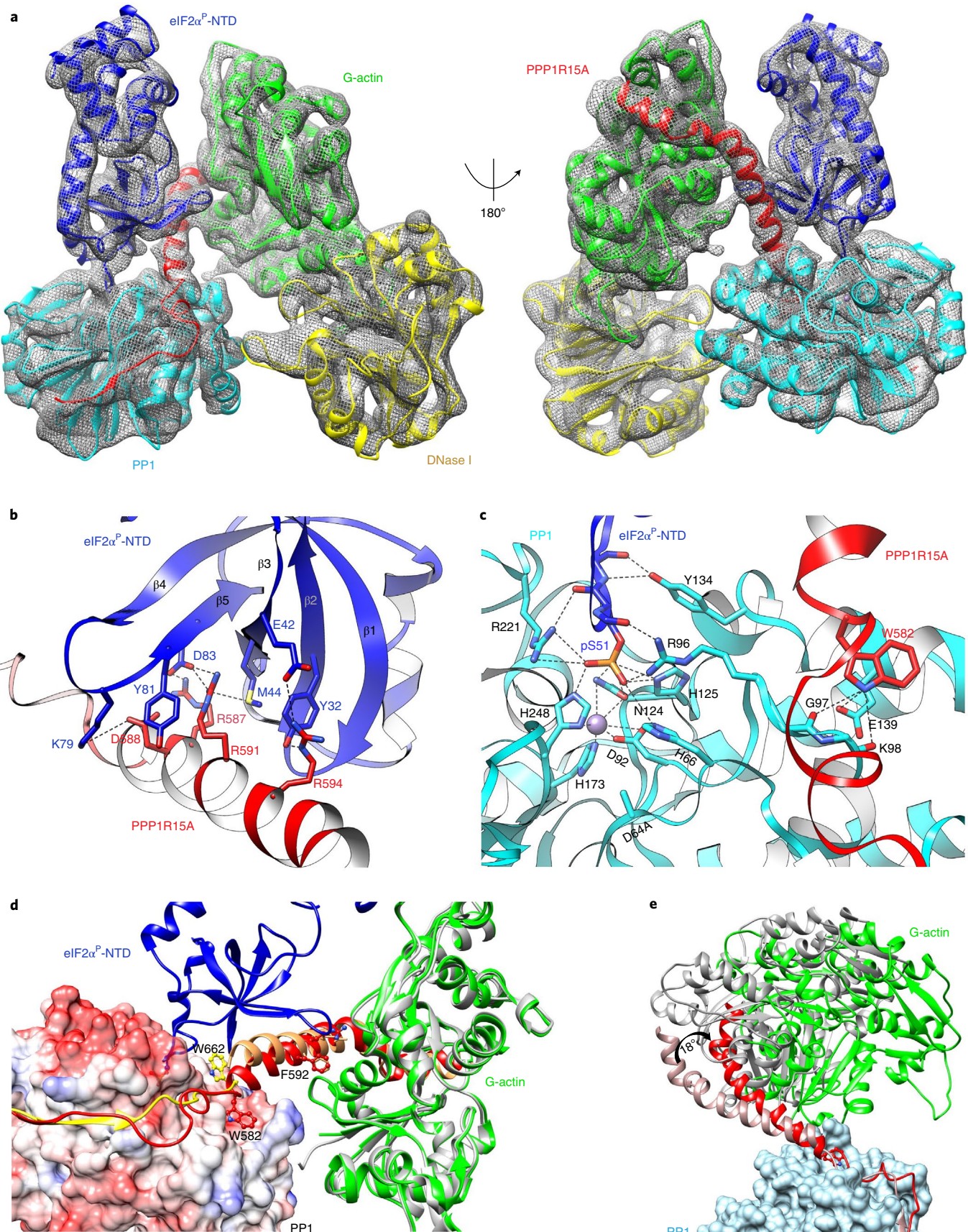

folded N-terminal domain of eIF2α, which extends flexibly from the eIF2 trimer, serves as a minimal specific substrate for kinases and phosphatases[40]. Cofactors other than G-actin may contribute to PPP1R15A-mediated dephosphorylation in cells. Interactions between the extended N termini of PPP1R15A (or B) and the β or γ subunits of the eIF2($α^P$) trimer (missing here) may contribute further to catalysis. For now, however, these are unproven. Furthermore, the biochemical properties of the minimal system studied here in vitro are adequate to explain the kinetics of eIF2$α^P$ dephosphorylation observed in cultured cells (Methods). Therefore, the enzyme described here is a valid starting point to study the dephosphorylation event that terminates signaling in the ISR.

The observation that higher-order contacts distant from both the substrate loop and enzyme active site promote dephosphorylation presents a challenge: whilst affinity of an enzyme for its substrate favors catalysis, residual affinity for the product is often anticatalytic. This conundrum could be settled by dephosphorylation-dependent structural changes to eIF2α that favor product dissociation. Allosteric changes have been noted in eIF2α upon Ser51 phosphorylation/dephosphorylation[36,38], but they do not extend to the region involved in the higher-order contacts identified here. Furthermore, given the similarity in contacts with the kinases that phosphorylate eIF2$α^0$ and the PPP1R15 holophosphatase that dephosphorylates eIF2$α^P$, a coherent allosteric change that promotes product release in both antagonistic reactions seems unlikely.

In an alternative scenario, cooperation between local contacts at the active site and higher-order contacts is important in formation of the pre-reaction complex. Enhanced binding of the holophosphatase to immobilized eIF2$α^P$, compared with its binding to nonphosphorylated eIF2α (in the BLI assay), speaks to this point. The PP1A$^{D64A}$ mutation, used to stall the reaction, destabilizes the M1 metal ion of the active site, which can no longer coordinate an eIF2α pSer51 oxygen[35]. Due to loss of these contacts the difference in affinity of the substrate (eIF2$α^P$) and product (eIF2$α^0$) for the wild-type holoenzyme may be greater than that reported by the BLI experiment; possible large enough to dominate the kinetics of product release. The higher-order contacts are clearly important, as both binding in the BLI assay and the 50-fold spread in catalytic efficiency between the apo and holophosphatase are sensitive to their disruption. However, it stands to reason that successful product dissociation requires that the affinity imparted by these distal contacts be tuned to differences in binding kinetics of the active site for the substrate versus the product. The higher 'off' rates of the product observed in BLI likely play into this. Similar considerations apply to eIF2α phosphorylation, in which higher-order contacts likely cooperate with contacts at the active site to stabilize a pre-phosphorylation complex[29,41]. We speculate that the relatively modest surface of eIF2α buried in both complexes (~1,200 Å$^2$)

reflects a constraint imposed on substrate and product affinity by catalytic efficiency. Similar considerations apply to the inherent competition between eIF2($α^P$) binding to eIF2B and its dephosphorylation by the PPP1R15 holophosphatases, as residues on eIF2α L$_{4,5}$ (residues 77–80) and β$_5$ (residues 81–85) are important to both interactions[36–38]. Here, the relatively high turnover of the eIF2B–eIF2$α^P$ complex[39] likely favors the one-way flow of substrate from eIF2B to the phosphatase.

PPP1R15's positive role in eIF2$α^P$ dephosphorylation appears limited to scaffolding the enzyme to favor the aforementioned higher-order contacts—PPP1R15A makes no contacts with the eIF2$α^P$ substrate loop. Nonetheless, local sculpting of the surface of PP1c by regulatory subunits is likely to affect eIF2$α^P$ dephosphorylation. For example, insertion of the eIF2$α^P$ substrate loop in the active site of PP1c is compatible with neither Phactr1, nor spinophilin, binding[5,7]. Thus, the process described here for selecting a globular domain of a phosphoprotein as substrate likely operates alongside the well-established mechanism for locally biasing access to the PP1c active site by regulatory subunits engaging its surface[9].

In eIF2α phosphorylation, formation of higher-order contacts with the substrate is dependent on allosteric coupling of back-to-back dimerization of the kinase to alignment of its eIF2α-binding helix G[42]. Precise alignment of components is also required for higher-order contacts in the tripartite holophosphatase. This fine tuning seems to have arisen solely by refinement of the PPP1R15 component, as features that enable it to function together with PP1c and G-actin are present in simpler eukaryotes that have no counterpart to PPP1R15. For example, mammalian actin Asp334 that contacts PPP1R15A Arg595 is conserved in yeast. Similarly conserved is the actin hydrophobic pocket that accommodates PPP1R15A Phe592. Indeed, PPP1R15A is functional as an eIF2$α^P$ phosphatase in yeast and its activity depends on the integrity of residues involved in what we now understand to be G-actin contacts[43]. It thus appears that in metazoans the holophosphatase was cobbled together from one rapidly evolving component (PPP1R15) and two off-the-shelf pre-existing blocks (PP1 and G-actin).

Actin is a highly dynamic protein and its partitioning between filamentous, F-actin, and monomeric, G-actin, is responsive to upstream signals. G-actin is a limiting ligand in some physiological reactions, subordinating them to the F/G-actin ratio[44]. Actin dynamics may have been co-opted to regulate the ISR in some circumstances, but these have yet to be identified. Alternatively, G-actin incorporation into the holophosphatase may have arisen simply by its availability as a convenient building block. The genetic evidence provided here for G-actin's role as a cofactor in eIF2$α^P$ dephosphorylation in cells renders these questions pertinent.

Potent inhibitors of PP1c exist, but they nonselectively block dephosphorylation. Mechanism-based pharmacological targeting

**Fig. 5 | Substrate-facing PPP1R15A mutations interfere with eIF2$α^P$ dephosphorylation. a**, Coomassie-stained SDS–PAGE PhosTag gels of eIF2α following dephosphorylation in vitro with PPP1R15A holophosphatases. Concentrations of the binary PPP1R15A–PP1A and G-actin are indicated, as are the genotypes of the PPP1R15A component. Shown is a representative of experiments reproduced at least three times. **b**, As in **a** but in the absence of G-actin. **c**, Display of the best fit of three independent experiments in panels **a** and **b** ±95% confidence intervals of the $k_{cat}/K_m$ of the indicated PPP1R15A–PP1A pairs in reactions lacking (apo) and containing (holo) G-actin. **d**, Two-dimensional flow cytometry plots of untreated and thapsigargin (Tg)-treated CHOP::GFP transgenic CHO-K1 cells transfected with mCherry alone or full-length (wild type or mutant) mouse PPP1R15A fused to mCherry at its C termini (as in Fig. 2e). **e**, Box (25th and 75th percentile) and whiskers plots (cut-off at the 99th percentile) of the CHOP::GFP signal in untreated and thapsigargin-treated cells from the gray region in **d** (cell numbers shown in **d**). **f**, Individual data points and mean ± s.d. of the ISR attenuation factor of the indicated subset of PPP1R15A::mCherry fusions from replicate experiments as in **d** (P values for two-tailed $t$-test comparisons shown, $n = 3$ independent experiments). **g**, Quantitation of stress granules in sodium arsenite-treated cells expressing the proteins as in **d** (all data, mean ± s.d., P values for two-tailed $t$-test comparisons shown, $n = 8$ biological replicates for mCherry, 16 for WT, 10 for W575 and 11 for R584A). Results are representative of experiments reproduced three times. **h**, Time-dependent change in the BLI signal arising from phosphorylated eIF2$α^P$ and nonphosphorylated eIF2$α^0$ immobilized on a BLI probe and introduced in a 4 µM solution of a ternary complexes (TC, PP1A$^{D64A}$–PPP1R15A–G-actin), binary complexes of the same lacking G-actin (BC) or G-actin alone. At 900 s into the association phase the probe was shifted to a solution lacking proteins, to record the dissociation phase. **i**, BLI signal arising from immobilized eIF2$α^P$ and ternary complexes of PP1A$^{D64A}$, G-actin and wild type or the indicated mutant PPP1R15A (as in **h**). Shown are representative traces of experiments reproduced at least three times. Source data for panels **a**–**c** and **e**–**i** are available online.

of substrate-specific holophosphatases is a long-sought goal, but we believe previous reports of success[45] to be in error[26,27]. Dependence of the eIF2α[P] dephosphorylation reaction on the precise alignment of the three components of the holophosphatase hints at a path towards this goal. As noted above, the catalytic and substrate-binding lobes

of the holophosphatase pivot about a hinge located between the PP1-facing and G-actin-facing portions of PPP1R15. In the catalytically favored conformation, the conserved PPP1R15A Trp582 may transition from its conventional pocket on the surface of PP1A (observed in the binary PP1A–PPP1R15B and PP1A–Phactr1

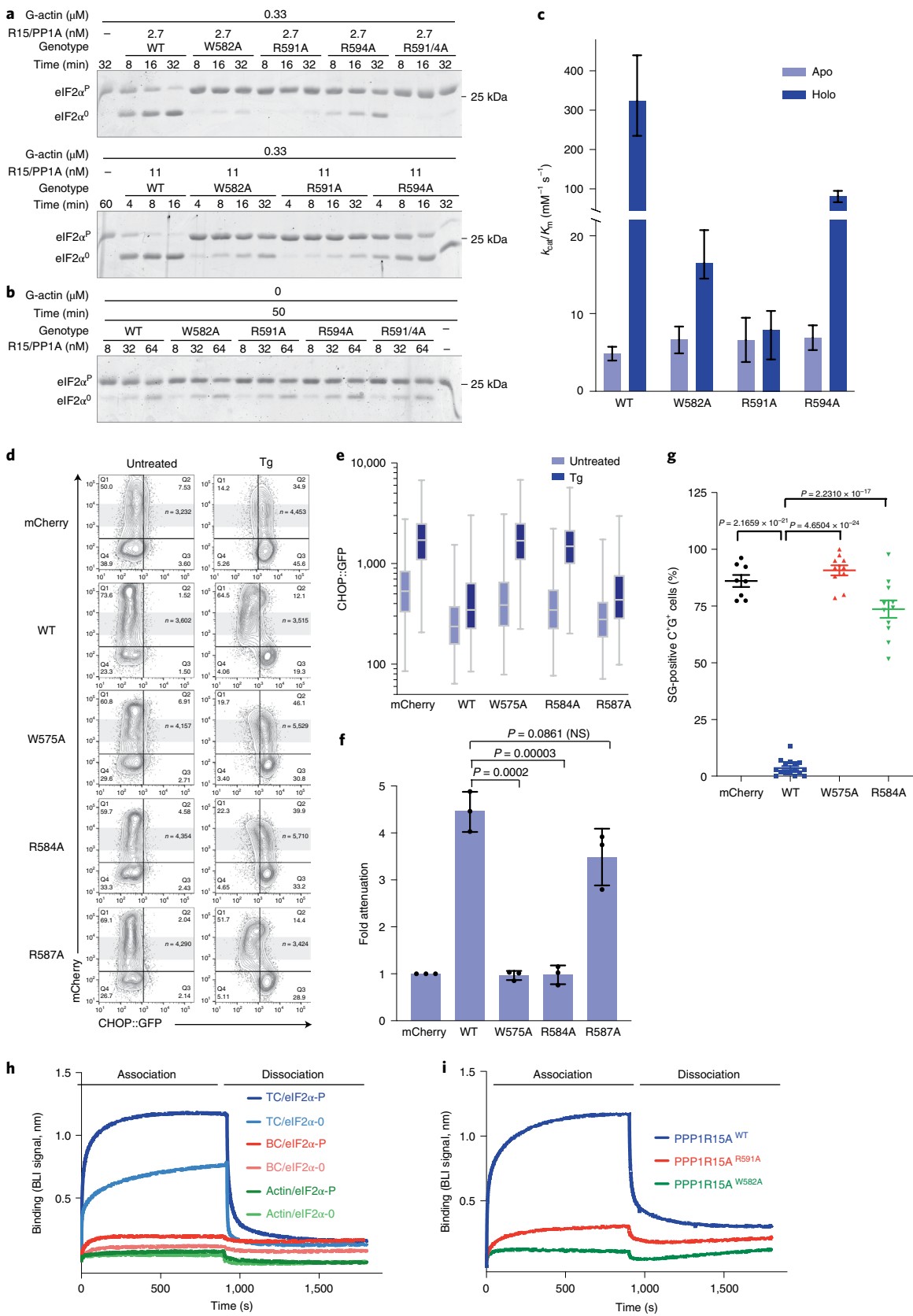

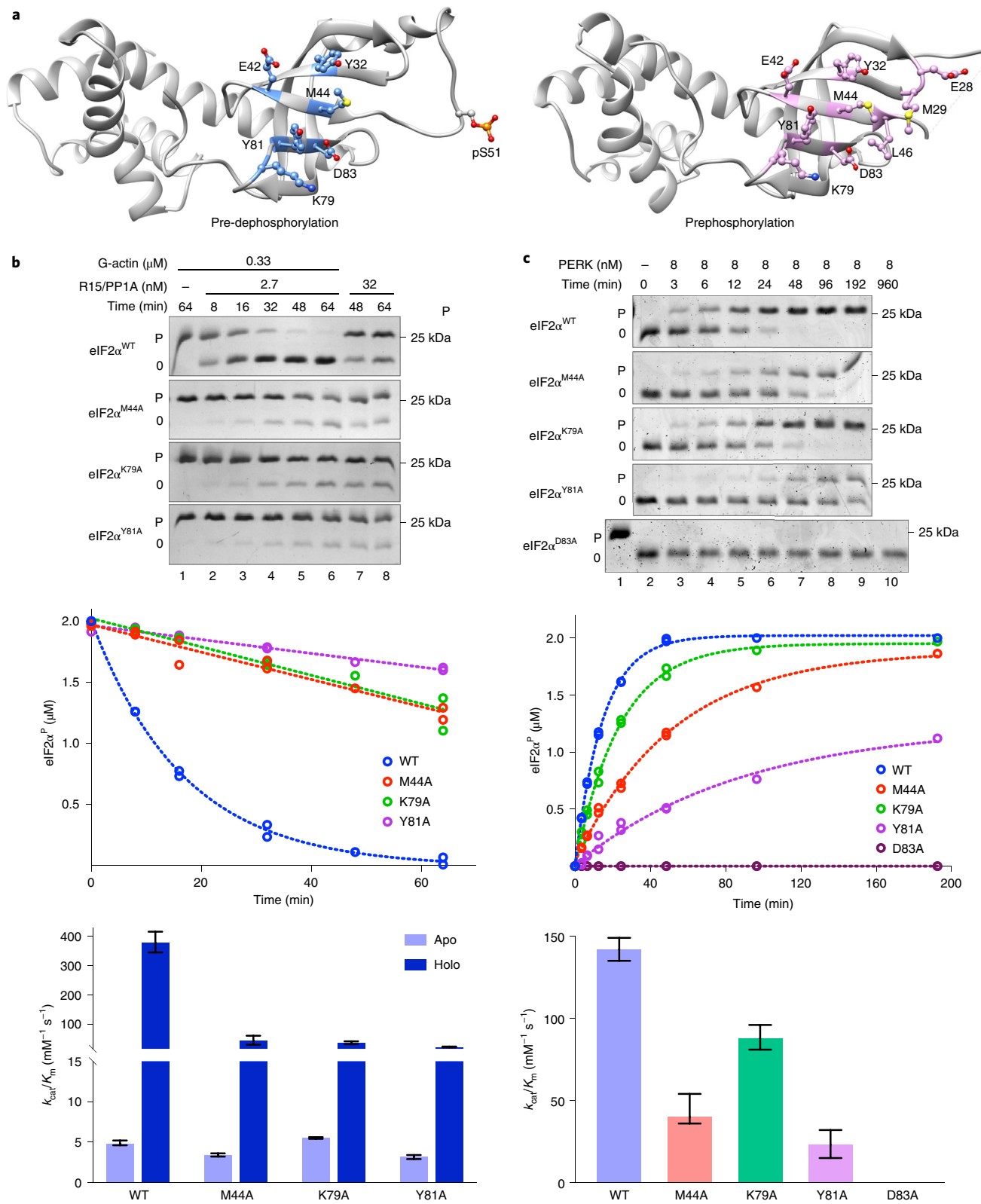

**Fig. 6 | Enzyme-facing substrate mutations interfere with eIF2α^P dephosphorylation. a**, Ribbon diagram of eIF2α from the pre-dephosphorylation complex (this study, left) and a pre-phosphorylation complex with PKR (PDB 2A19, right). Residues contacting the respective enzymes are highlighted. pSer51, shown on the right, was unresolved in PDB 2A19. **b**, Coomassie-stained PhosTag gels of wild type and the indicated eIF2α^P mutants (at 2 μM), following dephosphorylation in vitro with PPP1R15A holophosphatases (upper panel). Time-dependent progression of the dephosphorylation reactions above. The dotted line is fit to a first-order decay (of replicates) (middle panel). Best fit of three independent experiments in the 'upper panel' ±95% confidence intervals of the $k_{cat}/K_m$ of the indicated enzyme/substrate pairing in reactions lacking (apo) and containing (holo) G-actin (lower panel). Shown are representative example of experiments reproduced at least three times. **c**, As in **b** but reporting on the phosphorylation of the wild type and the indicated eIF2α^0 mutants (at 2 μM) by the indicated concentration of the kinase PERK. Source data for **b** and **c** are available online.

complexes[7,25]) to the novel pocket observed here. Ligands that bind the holophosphatase and stabilize the inactive conformation of Trp582 are predicted to selectively inhibit eIF2α[P] dephosphorylation by interfering with the higher-order contacts necessary for terminating signaling in the ISR. Time will tell if such insights, derived from these snapshots of the eIF2α[P] phosphatase in action, advance targeting of dephosphorylation reactions to beneficial ends.

## Online content

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

## Methods

**Protein expression and purification.** N-terminal His6-Smt3 human PPP1R15A[582–621] (UK2514) used for cocrystallization with G-actin and DNase I was expressed in *Escherichia coli* BL21 T7 Express *lysY/I*[q] cells (New England BioLabs, catalog no. C3013) and purified as previously described, with some modifications[46]. Briefly, the protein expression was induced by the addition of 0.5 mM isopropylthio-β-ᴅ-1-galactopyranoside (IPTG) for 16 h at 22 °C. Cell pellets were suspended in the HisTrap column-binding buffer (20 mM Tris–HCl pH 7.4, 0.5 M NaCl, 20 mM imidazole) containing protease inhibitors and 0.1 mg ml⁻¹ DNase I and 20 μg ml⁻¹ RNase A. The cells were crushed by a cell disruptor (Constant Systems) at 30 kPSI and the lysates were cleared by centrifugation for 1 h at 45,000*g*. The supernatant was applied to a pre-equilibrated 5 ml HisTrap column (GE Healthcare) and the bound protein was eluted with a 20–200 mM imidazole gradient using an FPLC purifier system (ÄKTA; GE Healthcare). Peak fractions were pooled and digested with SENP2 protease (at final concentration of 0.01 mg ml⁻¹; produced in-house) to cleave off the His6-Smt3 tag overnight at 4 °C. The remaining full-length fusion protein and His6-Smt3 tag were removed by binding back to a 5 ml HisTrap column. Following buffer exchange to lower salt buffer (10 mM HEPES–KOH pH 7.4, 50 mM NaCl), the target protein was further purified by cation exchange chromatography using a HiTrap SP HP column (GE Healthcare) and eluted by a 50–500 mM NaCl gradient in 10 mM HEPES–KOH pH 7.4. Target protein-containing peaks were collected and further purified using a HiLoad 16/600 Superdex 75 prep-grade gel filtration column equilibrated with 10 mM Tris–HCl pH 7.4 and 0.15 M NaCl. The elution peak fractions were pooled and ready to form a complex with G-actin/DNase I. Extra samples were snap-frozen in liquid nitrogen and stored at −80 °C.

His6-Smt3-tagged human PPP1R15A[553–621] with a C-terminal fused maltose-binding protein (MBP) tag and rabbit PP1A[D64A] (7–300, UK2720) were coexpressed using a bicistronic construct with a Shine–Dalgarno (SD) sequence inserted between the 3′ end of PPP1R15A and the 5′ end of PP1A coding sequences in *E. coli* BL21 T7 Express *lysY/I*[q] cells, induced as described above and purified by sequential Ni-NTA affinity chromatography, His6-Smt3 tag cleavage, reverse Ni-NTA affinity chromatography (to remove the His6-Smt3) and a HiLoad 16/600 Superdex 200 size-exclusion chromatography, as described above. MnCl₂ (1 mM) was included in all of the steps from IPTG induction. Sequential nickel chelating, His6-Smt3 tag cleavage and reverse nickel chelating were applied for making PP1A–PPP1R15A (UK2699), PP1A–PPP1R15A[R595E] (UK2750), PP1A–PPP1R15A[F592A;R595A] (UK2751), PP1A–PPP1R15A[R612A] (UK2752) and PP1A–PPP1R15A[F592A;R595E] (UK2755) for the dephosphorylation assay (Fig. 2a–c and Extended Data Fig. 2).

N-terminal His6-Smt3 human eIF2α[2–187] (eIF2α-NTD, UK2731) was purified by sequential nickel chelating, His6-Smt3 tag cleavage and reverse nickel chelating, as described above. The crude eIF2α-NTD sample was phosphorylated by incubating with GST-PERK (1:100 molar ratio) for 2 h at 37 °C in 100 mM NaCl, 25 mM Tris–HCl pH 7.4, 5 mM MgCl₂, 2.5 mM ATP and 1 mM TCEP. After phosphorylation, the sample was further purified by HiLoad 16/600 Superdex 75 size-exclusion chromatography equilibrated with 10 mM Tris–HCl pH 7.4, 0.15 M NaCl and 1 mM TCEP. Fractions were pooled and ready to form complexes with holophosphatase components.

Mutant derivatives of His6-Smt3 human eIF2α[2–187] (UK2849–52) were expressed as described above, but to compensate for their defective ability to serve as substrates of phosphorylation, the reaction was extended overnight at room temperature: substrate, at ~50 μM, was combined with purified GST-PERK at 10 nM in 100 mM NaCl, 25 mM Tris–HCl pH 7.4, 5 mM MgCl₂, 2.5 mM ATP and 1 mM TCEP and incubated for 16 h at room temperature. Completion of the reaction was verified by PhosTag SDS–PAGE and residual enzyme was removed by flowing the sample through a glutathione-sepharose resin.

Biotinylated, AviTagged human eIF2α[2–187] (UK2733, Fig. 5h,i)[39] was purified and phosphorylated as described above, and biotinylated AviTagged PPP1R15A[583–621] (wild type and mutants, UK2773, 6, 7, 9, Fig. 2d) recovered from cultures supplemented with biotin (100 μM).

Actin was purified from rabbit muscle[47], dialyzed against G buffer (5 mM Tris–HCl pH 8, 0.2 mM ATP, 0.5 mM DTT, 0.2 mM CaCl₂) for 3 days. The G-actin in the binary structure was further incubated with a ten-molar excess of latrunculin B (abcam, catalog no. ab14491). Partially purified (DP grade) bovine pancreatic DNase I powder was purchased from Worthington Biochemical (catalog no. LS002139). The DNase I powder was dissolved in 10 mM Tris–HCl pH 7.4, 0.05 M NaCl, 1 mM CaCl₂ and protease inhibitors before purified by a HiLoad 16/600 Superdex 75 prep-grade gel filtration column equilibrated with 10 mM Tris–HCl pH 7.4, 0.15 M NaCl and 1 mM CaCl₂. Peak fractions were pooled with addition of protease inhibitors. A 1:1.5 molar ratio of G-actin and DNase I was mixed before loading to a HiLoad 16/600 Superdex 75 prep-grade gel filtration column equilibrated with 10 mM Tris–HCl pH 7.4, 0.15 M NaCl and 1 mM CaCl₂, 0.2 mM ATP and 0.2 mM TCEP. The G-actin–DNase I complex peak fractions were collected and subsequently formed a complex with either human PPP1R15A[582–621] or PP1A[D64A]–PPP1R15A[553–624]–MBP/eIF2α-NTD[P]. The former complex was further purified by a HiLoad 16/600 Superdex 75 prep-grade gel filtration column equilibrated with 10 mM Tris–HCl pH 7.4, 0.15 M NaCl and 1 mM CaCl₂, 0.2 mM ATP and 0.2 mM TCEP. The latter complex was further purified by consecutive Superdex Increase 75 10/300 GL and Superdex Increase 200 10/300 GL gel

filtration columns equilibrated with 10 mM Tris–HCl pH 7.4, 0.15 M NaCl, 1 mM CaCl₂, 0.2 mM ATP, 0.2 mM TCEP and 1 mM MnCl₂. The G-actin–DNase I–PPP1R15A[582–621] complex peak fractions were concentrated to 10 mg ml⁻¹ for crystallization; the PP1A[D64A]–PPP1R15A[553–624]-MBP–eIF2α[P]-NTD–G-actin–DNase I complex was concentrated to 5 mg ml⁻¹ for making cryo-EM grids.

**Structural analysis.** *Crystallography.* Equal volume (100 nl) of PPP1R15A[582–621] –G-actin–DNase I complex at 10 mg ml⁻¹ and reservoir solutions were dispensed to each drop of a 96-well sitting-drop crystallization tray for broad screening, and the crystallization took place at 20 °C. The initial, thin, plate crystals grew at 8% PEG 4000 and 0.1 M sodium acetate pH 4.6. The best dataset was collected from a crystal in 10% PEG 4000 and 0.1 M sodium acetate pH 5 microseeded with crushed initial crystals. Diffraction data were collected at beamline i03 at the Diamond Synchrotron Light Source (DLS) at a wavelength of 0.976 Å and temperature of 100 K using DLS-developed generic data acquisition software (GDA v.9.2). Data were processed by the XIA2 pipeline implementing DIALS[48] for indexing and integration, Pointless[49] for space group determination and Aimless[50] for scaling and merging. The structure was solved by molecular replacement using Phaser[51] and two copies of G-actin–DNase I (PDB 2A42) were found in an asymmetric unit. PPP1R15A[582–621] was manually built to the difference map in Coot (v.0.9.2)[52]. Further refinement was performed iteratively using Coot, phenix.refine[53] and REFMAC5 (ref. [54]). These structural data analysis programs were from the CCP4i (v.7.1.012)[55] and Phenix (v.1.18.2-3874)[56] suites. MolProbity[57] was consulted for structure validation with 97.9% Ramachandran favored region and no Ramachandran outliers (Table 1). Molecular graphics were generated using UCSF Chimera (v.1.14)[58] and PyMOL (v.1.3; Schrödinger, LLC).

*Cryo-EM.* UltrAuFoil 0.6/1 300-mesh grids (Quantifoil) were glow discharged in residual air at 25 mA for 3 min using a Pelco EasiGLOW. A 5 μl portion of the PP1A[D64A]–PPP1R15A[553–624]-MBP/eIF2α-NTD[P]–G-actin–DNase I complex at 5 mg ml⁻¹ was mixed with 0.2 μl of 5.6 mM Triton X-100 immediately before plunging. The additional ~0.22 mM Triton X-100 improved particle orientation distribution notably. A 3.5-μl sample was deposited to grids and fast-frozen in liquid ethane cooled by liquid nitrogen using a Vitrobot Mark IV (ThermoFisher). Grids were blotted for 3.5 s with a force at −7 in a 100% humidity chamber at 4 °C. Movie stacks were collected on a Titan Krios operated at 300 keV equipped with a K3 camera (Gatan) at the super-resolution counting mode using EPU software (v.2). Images were recorded at ×130,000 magnification corresponding to 0.652 Å per pixel (counted super-resolution 0.326 Å per pixel) using a 20-eV energy filter. Image stacks have 49 frames for an accumulated dose of ~50 e⁻/Å² in a total exposure time of 1.3 s. The defocus range was from −2.8 μm to −1 μm. A total of 4,025 micrographs were collected from a single grid.

WARP (v.1.0.6)[59] was used for motion correction and CTF estimation, during which the super-resolution data were binned to given a pixel size of 0.652 Å. Particles were automatically picked using the BoxNet algorithm of WARP. A total of 132,495 particles stacks generated by WARP were imported to cryoSPARC (v.3.1)[60]. Four initial models were generated by initial three-dimensional (3D) reconstruction from scratch, followed by a heterogeneous refinement with the four initial models and the whole particle set as input. Model 1 (29,879 particles) represented the G-actin–DNase I complex; models 2 and 3 were ice or unknown junk and model 4 (76,280 particles) represented the full complex. Another two rounds of ab initio 3D reconstruction and heterogeneous refinement of the model 4 particle stacks were performed to remove extra bad particles, resulting in 60,413 final particles for the full complex. Further 3D classification of these 60,413 particles generated two similar classes with worse resolutions at the core region, which had noticeable differences only at the intensities of the flexible MBP tag density. Nonuniform refinement[61] was performed to improve the resolution, resulting in a 3.96-Å resolution map with corrected autotightened mask at a gold standard FSC of 0.143. Defocus, global CTF and local refinement did not improve the resolution. A ResLog plot[62] of resolutions against particles reached a plateau with the final particle set.

Docking of PP1c (PDB 4MOV), G-actin–DNase I (PDB 2A42) and eIF2α-NTD (PDB 1KL9) to the cryo-EM map was performed in UCSF Chimera. Coot was used for further local model building. A ResolveCryoEM map[63] was consulted to build regions with better quality, such as the C-terminal helical domain of PPP1R15A, the pSer 51 loop and the β-barrel of eIF2α[P]-NTD. The atomic model was refined with Phenix real-space refinement[53] using the ResolveCryoEM map. MolProbity was used for structural validation (Table 2). UCSF Chimera was used to prepare graphic figures. The angular distribution was generated by the final nonuniform refinement in cryoSPARC and converted by UCSF pyem[64] (v.0.5) to plot in UCSF Chimera.

**Enzymatic activity.** Measuring eIF2α[P] dephosphorylation rates. Dephosphorylation reactions followed established procedures[25]: binary complexes of wild type or the indicated mutants of human PPP1R15A[553–621] (fused to *E. coli* MBP at their C termini, as a solubility tag) with associated untagged rabbit PP1A[7–300], purified from *E. coli* as described above and maintained as concentrated stocks (≥10 μM), were quickly diluted into assay buffer (100 mM NaCl, 25 mM Tris–HCl pH 7.4), 200 μM MnCl₂, 1 mM TCEP, 5% glycerol, 0.02% Triton X-100) to

the indicated concentrations (2–100 nM) in a 200-μl PCR tube maintained at 20 °C. G-actin was added (to a final concentration 330 nM, unless indicated otherwise) and the sample was incubated at 20 °C for 5 min to allow ternary complex assembly. The reaction was initiated by adding phosphorylated human eIF2α$^{2-187}$ (purified from *E. coli* and phosphorylated in vitro with GST-PERK, as indicated above) to a final concentration of 2 μM.

Samples were removed from the reaction at intervals, quenched into sample buffer (62.5 mM Tris–glycine pH 6.8, 50 mM DTT, 2% SDS, 10% glycerol), loaded onto a 10 cm × 0.1 cm 15% SDS–PAGE gel with 50 μM PhosTag reagent and 100 μM MnCl$_2$, resolved at 200 V for 80 min, stained with Coomassie and fluorescently scanned on a Licor Odyssey using the built-in 680-nm laser.

The intensity of the phosphorylated and nonphosphorylated species in each lane was quantified using NIH image[65].

*Measuring eIF2α phosphorylation rates.* Human eIF2α$^{2-187}$, final concentration 2 μM, and GST-PERK, final concentration 8 nM (both purified from *E. coli*) in kinase buffer (100 mM NaCl, 25 mM Tris–HCl pH 7.4, 5 mM MgCl$_2$, 2.5 mM ATP, 1 mM TCEP, 5% glycerol, 0.02% Triton X-100), were combined in a PCR tube at 20 °C and reaction progress was monitored by SDS–PAGE PhosTag electrophoresis and Coomassie stain, as described above.

**Biolayer interferometry.** BLI experiments were conducted at 30 °C on the FortéBio Octet RED96 system, at an orbital shake speed of 600 r.p.m., using streptavidin (SA)-coated biosensors (Pall FortéBio) in an assay buffer of 100 mM NaCl, 25 mM Tris–HCl pH 7.4, 200 μM MnCl$_2$, 1 mM TCEP, 5% glycerol, 0.02% Triton X-100.

Biotinylated ligands, either wild type or mutant PPP1R15A$^{583-621}$ (Fig. 2d) or phosphorylated or nonphosphorylated eIF2α$^{2-187}$ (Fig. 5h,i) both expressed in and purified from *E. coli* as C-terminally AviTag-His6-tagged proteins were loaded onto biosensor at a concentration of 150 nM to a binding signal of 1–2 nm, followed by baseline equilibration in buffer. Association reactions with analyte: G-actin (Fig. 2d) or wild-type or mutant complexes of PPP1R15A$^{553-624}$ (fused to MBP at its C terminus) with PP1A$^{D64A}$, with or without G-actin (Fig. 5h,i), in the assay buffer described above, were conducted with a reaction volume of 200 μl in 96-well microplates (Greiner Bio-One).

**ISR activity in cells.** *Cell culture and transfection.* CHO-K1 wild type (stress granule experiments) or C30 CHOP::GFP[15] (flow cytometry experiments) cells were maintained in Ham's F12 medium supplemented with 10% fetalclone II serum (Hyclone) and 1× penicillin–streptomycin. For flow cytometry, cells were seeded at a density of 1.5 × 10$^5$ cells per well in 12 well dishes, and 24 h later were cotransfected with 50 ng mCherry, R15A-mCherry wild type or mutant expression vector and 950 ng of BFP marked empty or actin (WT or E334R) expression vector (1:19 ratio of R15::actin) using lipofectamine LTX (Life Technologies) at a ratio of 3 μl of lipofectamine LTX and 1 μl of Plus reagent per ng of DNA in the subsequent complex formation in OPT-MEM according to the manufacturer's instructions. At 12 h post-transfection the medium was changed and thapsigargin (Calbiochem) added at 200 nM, where indicated. The cells were washed and released from the dishes in PBS–4mM EDTA with 0.5% BSA, and 1 × 10$^4$ BFP$^+$ cells were subject to flow cytometry analysis[25], applying the gating strategy described in Extended Data Fig. 5. The flow cytometry data were acquired on a BD flow cytometer using FACSDIVA (v.8.0.1, BD Bioscience) and analyzed using FlowJo v.8.0, NIH Fiji v.1.0 and Excel2016.

Experiments tracking formation of stress granules were adapted from ref. [66]: CHO-K1 cells were transfected on poly(lysine)-coated multiwell slides. DNA complexes composed of 100 ng each of G3BP-GFP and R15A-mCherry expression vector, and 450 ng of empty plasmid carrier, were formed in a ratio of 3 μl ng$^{-1}$ DNA and 1 μl of Plus reagent per ng of DNA in the subsequent complex formation in OPT-MEM according to the manufacturer's instructions. After 15 min the complexes were diluted into 0.5 ml CHO medium and 125 μl was aliquoted into four wells containing cells. Medium was changed at 16 h post-transfection and 4 h later was replaced with medium containing 0.5 mM sodium arsenite. Cells were then washed and fixed in PBS–4% paraformaldehyde, followed by washing and mounting for microscopy. Micrographs of the red and green channels were taken on an EVOS (M5000 1.0.466.664) inverted photomicroscope using the ×20 magnification lens.

**Quantification and statistical analysis.** *Enzyme kinetics.* At concentrations well below the substrate $K_m$ the conversion to product (that is, substrate consumption) in an enzymatic reaction follows first-order kinetics and can be described ideally as a monophasic exponential decay.

When [S] ≪ $K_m$, enzymatic reactions are first order (their rate is proportional to [S]). This follows from the Michaelis–Menten equation:

$$v = \frac{d[P]}{dt} = -\frac{d[S]}{dt} = V_{max}\frac{[S]}{K_m + [S]} = k_{cat}[ENZ]\frac{[S]}{K_m + [S]}$$

If [S] ≪ $K_m$ then this rearranges to:

$$v = -\frac{d[S]}{dt} = k_{cat}[ENZ]\frac{[S]}{K_m}$$

Solved for $k_{cat}/K_m$ this rearranges to:

$$\frac{k_{cat}}{K_m} = \frac{\frac{d[S]}{dt}}{[S][ENZ]}$$

The experimental $k_{obs}$ of the conversion of substrate to product (fit to a monophasic exponential decay) corresponds to the term:

$$(d[S]/dt)/[S]$$

$k_{cat}/K_m$ can therefore be extracted from the experimentally determined $k_{obs}$ and the known concentration of enzyme [ENZ]:

$$\frac{k_{cat}}{K_m} = \frac{k_{obs}}{[ENZ]}$$

At 2 μM, eIF2α is well below the substrate $K_m$ of both the PPP1R15 holophosphatase and GST-PERK[25]. Therefore, the time-dependent substrate depletion (S) in the phosphorylation and dephosphorylation reactions was fitted to a one-phase decay function in GraphPad Prism v.9 using the model below:

$$S = (S_0 - S_{Plateau})\exp(-Kt) + S_{Plateau}$$

($S_0$ was constrained to the concentration of substrate at $t = 0$, $S_{Plateau}$ was set to zero, as at $t = \infty$, $S = 0$).

Due to practical limitations in the time range over which one can extend the dephosphorylation assay, or accurately obtain soundings of its progression, the range over which we can measure $k_{obs}$ is limited. To measure reaction progression accurately with the faster holophosphatase(s) and the much slower apoenzymes, we designed the assays to have similar $k_{obs}$ by varying [ENZ]: in assays of the various holophosphatases [ENZ] = 2 nM, whereas in assays of the apoenzyme [ENZ] was ~25 times higher. Therefore, $k_{cat}/K_m$ (which is simply $k_{obs}/[ENZ]$) emerges as a useful metric to compare catalytic efficiency of the holo and apoenzymes measured in assays that have very different [ENZ]. However, as the experiments most critical to the conclusions drawn here—namely, those comparing dephosphorylation by different holophosphatases—were conducted with the same enzyme concentrations [ENZ] and the same initial substrate concentrations [S]$_0$, and as $k_{cat}/K_m = k_{obs}/[ENZ]$, the fold difference in $k_{cat}/K_m$ of such reactions (reported in Figs. 2c, 5c and 6b,c) is identical to the fold difference of their directly measured $k_{obs}$.

*Computational analysis of in vivo versus in vitro kinetics of eIF2α$^P$ dephosphorylation.* Following inactivation of the cognate eIF2α kinase, the eIF2α$^P$ signal in the immunoblot of CHO-K1 cell lysates decays exponentially, with an observed rate constant ($k_{obs}$) of 0.0013 s$^{-1}$ (ref. [28]) to 0.0039 s$^{-1}$ (ref. [27]). Modeling the time-dependent change in the eIF2α$^P$ signal (that is, the dephosphorylation process) to a first-order decay is justified by the observation that the concentration of eIF2α in CHO-K1 cells, ~1 μM, is well below the $K_m$ of the holophosphatase for its substrate, >20 μM (refs. [25,67]). Working at the catalytic efficiency of the PP1A–PPP1R15A–G-actin holoenzyme observed here in vitro ($k_{cat}/K_m \sim 0.4 \times 10^6$ s$^{-1}$ M), a similar enzyme present in the cell at a concentration of 4–10 nM could account for the $k_{obs}$ of eIF2α$^P$ dephosphorylation.

Cellular concentrations of PP1A vary by tissue (70 nM–3 μM)[68], but at the median concentration (435 nM) recruitment of just 2.5% of that pool into a PPP1R15A-containing complex would account for the observed rate of eIF2α$^P$ dephosphorylation in cells.

These crude estimates suggest that, at reasonable concentrations, an enzyme consisting of the conserved C-terminal portion of PPP1R15A could account for the activity observed in cells.

*FACS (ISR quantitation).* As the effector protein in these assays is a PPP1R15A::mCherry fusion, the intensity of the fluorescence in the red channel (the *y* axis of the two-dimensional FACS scan) reports on the level of effector protein in each cell and facilitates gating on populations expressing similar levels of effector in comparing the median and standard error of the CHOP::GFP signal (ISR channel).

The ISR attenuation factor of wild-type and mutant PPP1R15A proteins (used in Figs. 2g, 3c and 5f) was derived as follows:

$$F_{attenuation} = \left(\bar{GFP}^{TG}_{mCherry}\right)/\left(\bar{GFP}^{TG}_{PPP1R15A}\right)$$

$\bar{GFP}$, mean of the median GFP fluorescence in replicate experiments; TG, thapsigargin-treated cells; mCherry, expressing mCherry alone; PPP1R15A, expressing the cognate wild-type or mutant PPP1R15A:: mCherry fusion protein.

Because it does not take into account the effect of the expressed PPP1R15A on the basal level of the ISR marker, this method of calculation underestimates the attenuation of the ISR, especially by the more potent PPP1R15A derivatives. However, as this tends to underestimate differences in the effect of various PPP1R15A derivatives on the ISR, it strengthens the reliability of such differences when detected.

*Quantification of stress granules.* Micrographs were imported into NIH-Fuji and transfected cells expressing both mCherry and G3BP-GFP were marked as double positive and counted using the cell counter plugin. These same cells were then assigned to stress granule-positive (two or more granules) or stress granule-negative (no detectable granules) and marked and counted. The counts for each micrograph were exported into Excel and the percentage of stress granule-positive cells over the total number of double-positive transfected cells was calculated. A total of 8–16 images containing an average of greater than 50 transfected cells were counted for each condition. A representative of three experiments is shown.

**Reporting Summary.** Further information on research design is available in the Nature Research Reporting Summary linked to this article.

## Data availability

The atomic coordinates and structure factors of the crystal structure of the DNase I–G-actin–PPP1R15A$^{582-621}$ complex have been deposited to the PDB with accession code 7NXV. Electron microscope density maps and atomic models of the pre-dephosphorylation complex have been deposited in the EMDB and PDB, respectively, with accession codes EMD-12665 and PDB 7NZM. Structures under PDB accession codes 4MOV, 2A42 and 1KL9 were used as initial models for refinement. Other structures used as alignment for illustration are available in PDB, including 3EKS, 3J8A, 2A19, 6I3M, 6K71, 7D45, 3JAP,1Q46, 6K72, 3CW2 and 3JAP. Source data are provided with this paper.

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

## Acknowledgements

We thank M. Bollen (KU Leuven) for suggesting the PP1A$^{D64A}$ trap, V. K. Dickson, S. Marciniak, J. Chambers, A. F. Zyryanova and L. Neidhardt (CIMR), A. Crespillo-Casado (MRC-LMB) and Y. Wong (Calico) for critical comments and advice. Diamond Light Source i03 (mx21426) for X-ray crystal structure data collection. S. Hardwick, D. Chirgadze and L. Cooper (Cryo-EM facility, Department of Biochemistry, University of Cambridge) for help with Cryo-EM data collection and processing. The CIMR core flow cytometry facility for help with cell sorting. The Huntington laboratory (CIMR) for access to the BLI Octet machine. This research was funded in part by the Wellcome Trust (200848/Z/16/Z) and Calico Life Sciences LLC. For the purpose of open access, the author has applied a CC BY public copyright licence to any author-accepted manuscript version arising from this submission.

## Author contributions

D.R., Y.Y. and H.P.H. conceived the project, designed the experiments, analyzed the data, prepared figures and tables and co-wrote the manuscript. Y.Y. expressed and purified the proteins used here and conducted all the structural work. H.P.H. conducted all cell-based and some binding experiments. D.R. did most of the enzymology.

## Competing interests

The authors declare no competing interests.

## Additional information

**Extended data** is available for this paper at https://doi.org/10.1038/s41594-021-00666-7.

**Correspondence and requests for materials** should be addressed to David Ron.

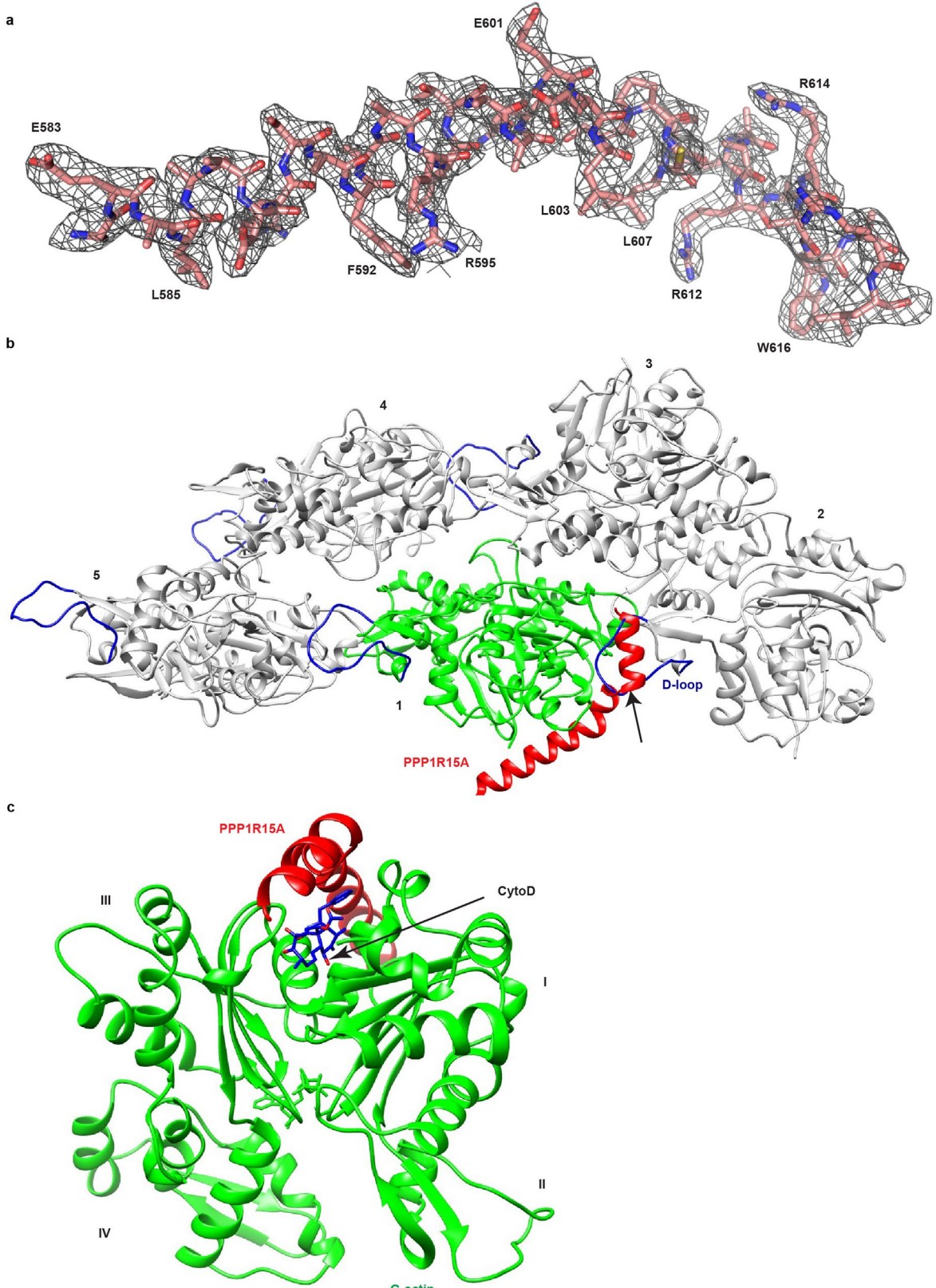

**Extended Data Fig. 1 | PPP1R15A binding to G-actin is exclusive to F-actin formation and CytoD binding.** (a) Stick diagram of PPP1R15A and the corresponding 2Fo-Fc map, contoured at 1.0σ within 2.0 Å of PPP1R15A atoms. (b) Ribbon diagram of F-actin (PDB 3J8A, tropomyosin omitted) in grey with protomers numbered. The PPP1R15A/G-actin complex (in red and green respectively, with DNase I removed) has been aligned with protomer 1. Actin D-loops are shown in blue and the clash between D-loop insertion and PPP1R15A binding to G-actin is indicated by the arrow. (c) The PPP1R15A/G-actin complex with actin domains numbered and a docked molecule of cytochalasin D (from PDB 3EKS), clashing with the C-terminal helix of PPP1R15A.

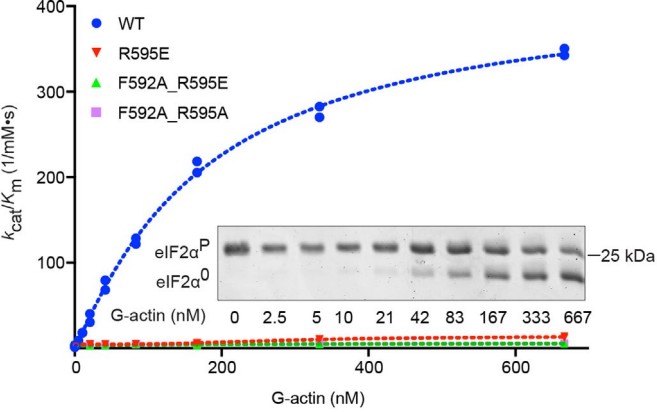

**Extended Data Fig. 2 | G-actin-facing PPP1R15A mutations interfere with eIF2αP dephosphorylation.** Plot of the $k_{cat}/K_M$ of holophosphatases comprised of indicated wildtype and mutant PPP1R15A paired with PP1A as a function of G-actin concentration. A PhosTag gel (as in Fig. 2a) of a representative experiment with the WT enzyme is shown in the inset.

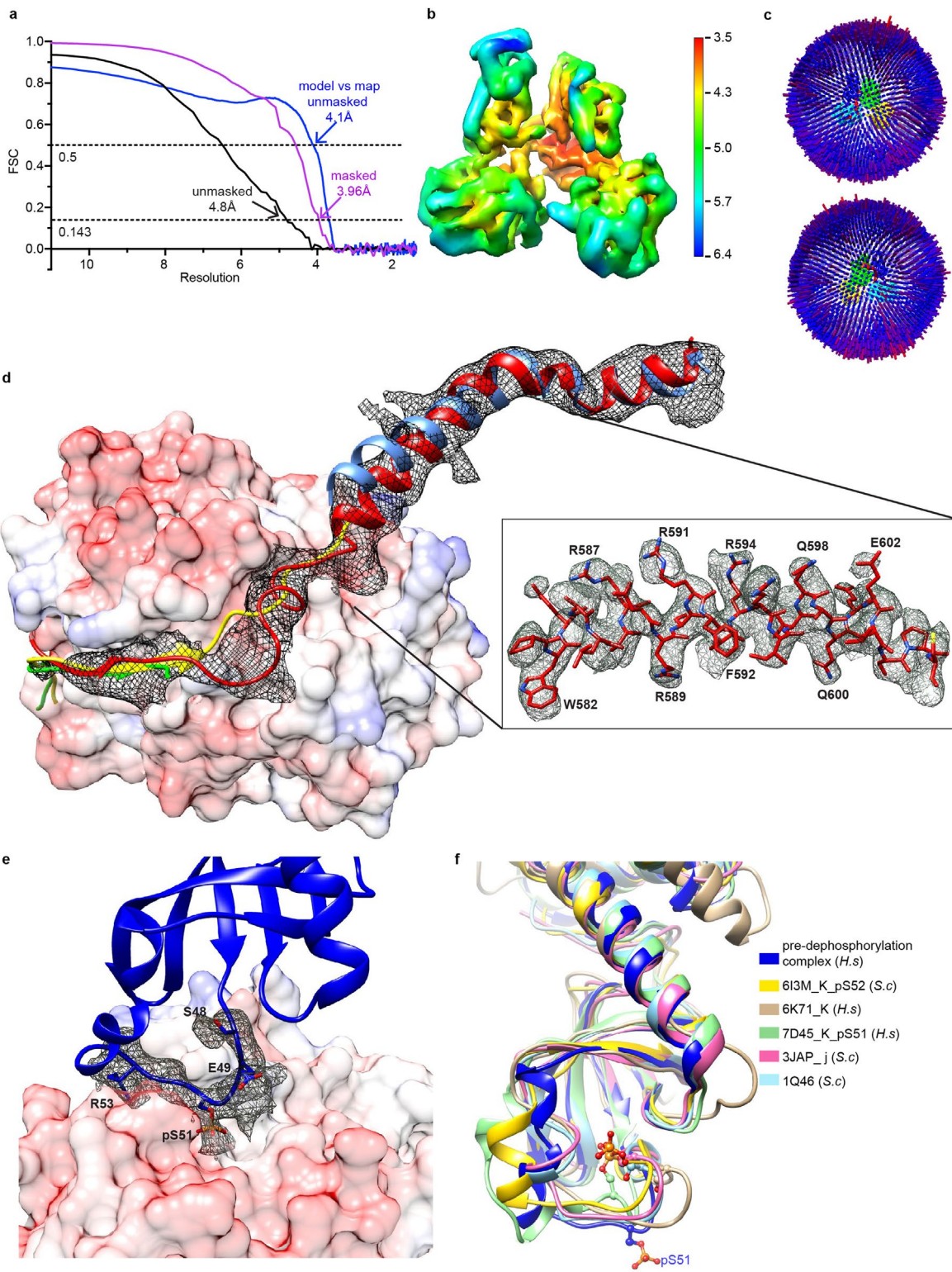

**Extended Data Fig. 3 | See next page for caption.**

**Extended Data Fig. 3 | Further about the cryo-EM structure of the eIF2αP pre-dephosphorylation complex.** (**a**) The Fourier shell correlation (FSC) curves for masked (pink, produced using cryoSPARC), unmasked (black, produced using cryoSPARC) maps, and the curve for the unmasked model and map correlation (blue, produced in Phenix.real_space_refine) are shown. The resolutions at which FSC for masked map drops below 0.143 and model map correlation drops below 0.5 are indicated. (**b**) The 3D density map from non-uniform refinement in cryoSPARC is depicted by local resolution on a colour scale shown on the right. (**c**) Angular distribution plots of particles that contributed to the final map are viewed from the front and the back of the density map. The number of particles with respective orientations are represented by length and coloured cylinders (from blue to red). In the centre, the cryo-EM map of the pre-dephosphorylation complex is coloured and angled the same as in Fig. 4a. (**d**) Superposition of the GS-FSC density map for PPP1R15A in the cryo-EM pre-dephosphorylation complex (red), PPP1R15A from the crystal structure of its binary complex with PP1A (PDB 4XPN, green) and PPP1R15B from the crystal structure of its binary complex with PP1G (PDB 4V0X, yellow) (both aligned by PP1c) and PPP1R15A from the crystal structure of its binary complex with G-actin/DNase I (here, light blue, aligned by its G-actin). Inset is the density modified map of PPP1R15A[581–606] from the Cryo-EM pre-dephosphorylation complex. (**e**) Density modified map for the eIF2α[P] substrate loop (residues 48-53) on the surface of PP1A, from the cryo-EM structure (eIF2α Arg52 sidechain was not modelled as there is no corresponding density and eIF2α Leu50 sidechain is omitted for clarity). (f) Alignment of human (*H.s*) eIF2α Ser51 loop from the pre-dephosphorylation complex with the structure of isolated yeast (*S.c*) eIF2α (PDB 1Q46), eIF2α[P] from human (*H.s*) (PDB 7D45 and 6K71) and yeast (PDB 6I3M) eIF2(αP)/eIF2B complexes and eIF2α from the yeast 48 S preinitiation complex (PDB 3JAP). PDB entry, chain and species are indicated. Note the unusual outward-facing orientation of phosphorylated Ser51 (pS51) from the pre-dephosphorylation complex.

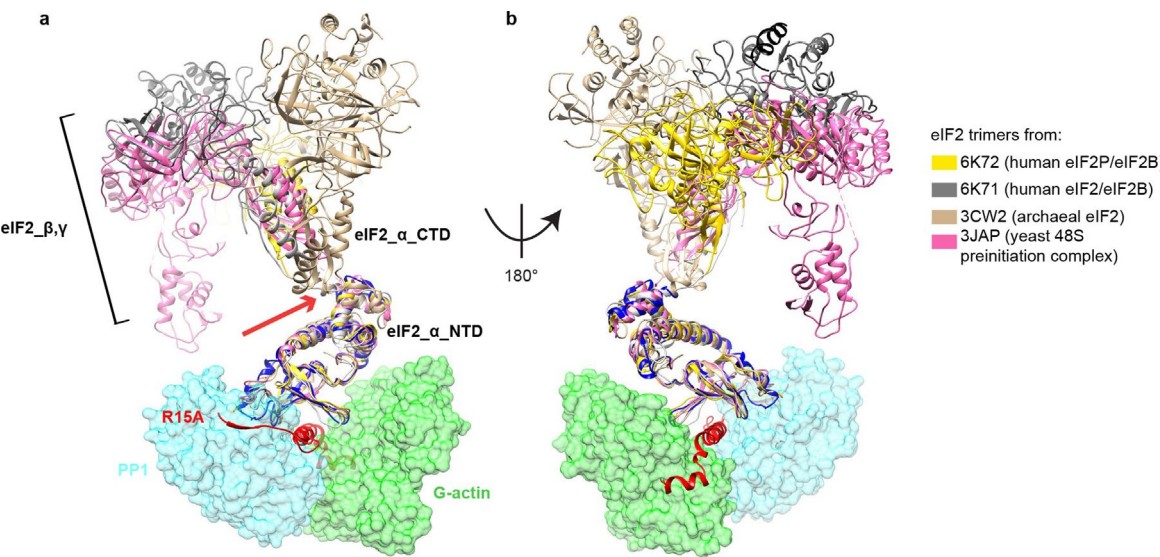

eIF2 trimers from:
- ■ 6K72 (human eIF2P/eIF2B)
- ■ 6K71 (human eIF2/eIF2B)
- ■ 3CW2 (archaeal eIF2)
- ■ 3JAP (yeast 48S preinitiation complex)

**Extended Data Fig. 4 | Alignment of the eIF2 trimer to the pre-dephosphorylation complex.** Alignment of the eIF2 trimer to the pre-dephosphorylation complex (coloured as in Fig. 4a). Note that a complete eIF2 trimer can fit well in the core holophosphatase. A full-length PPP1R15A could also be extended from the N-terminus of the fragment present in the complex here, with possible contacts with eIF2 trimer. The flexible hinge connecting the N-terminal domain of eIF2α (eIF2_α_NTD) and C-terminal domain of eIF2α (eIF2_α_CTD), which accounts for diversity in dispositions of the attached eIF2β and γ subunits, is indicated by a red arrow. The PDB accession codes are noted.

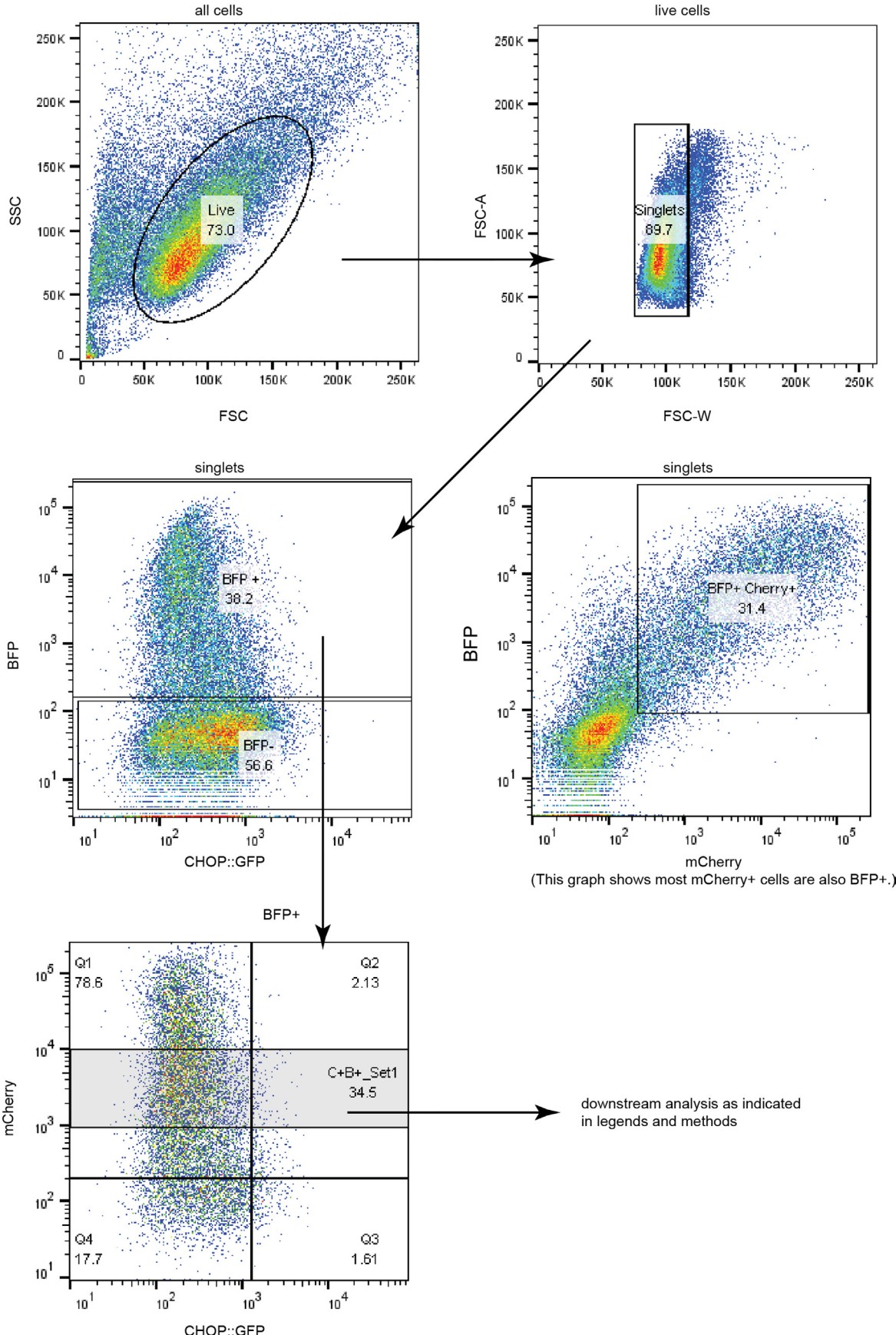

**Extended Data Fig. 5 | The gating strategy for flow cytometry experiments.** Shown is the gating strategy for flow cytometry experiments displayed in Figs. 2, 3 and 5. The percentage of cells from the proceeding gate is shown for each subsequent gate.

# nature research

# Reporting Summary

Nature Research wishes to improve the reproducibility of the work that we publish. This form provides structure for consistency and transparency in reporting. For further information on Nature Research policies, see our Editorial Policies and the Editorial Policy Checklist.

## Statistics

For all statistical analyses, confirm that the following items are present in the figure legend, table legend, main text, or Methods section.

| n/a | Confirmed | |
|---|---|---|
| ☐ | ☒ | The exact sample size ($n$) for each experimental group/condition, given as a discrete number and unit of measurement |
| ☒ | ☐ | A statement on whether measurements were taken from distinct samples or whether the same sample was measured repeatedly |
| ☐ | ☒ | The statistical test(s) used AND whether they are one- or two-sided *Only common tests should be described solely by name; describe more complex techniques in the Methods section.* |
| ☒ | ☐ | A description of all covariates tested |
| ☒ | ☐ | A description of any assumptions or corrections, such as tests of normality and adjustment for multiple comparisons |
| ☐ | ☒ | A full description of the statistical parameters including central tendency (e.g. means) or other basic estimates (e.g. regression coefficient) AND variation (e.g. standard deviation) or associated estimates of uncertainty (e.g. confidence intervals) |
| ☐ | ☒ | For null hypothesis testing, the test statistic (e.g. $F$, $t$, $r$) with confidence intervals, effect sizes, degrees of freedom and $P$ value noted *Give P values as exact values whenever suitable.* |
| ☒ | ☐ | For Bayesian analysis, information on the choice of priors and Markov chain Monte Carlo settings |
| ☒ | ☐ | For hierarchical and complex designs, identification of the appropriate level for tests and full reporting of outcomes |
| ☒ | ☐ | Estimates of effect sizes (e.g. Cohen's $d$, Pearson's $r$), indicating how they were calculated |

*Our web collection on statistics for biologists contains articles on many of the points above.*

## Software and code

Policy information about availability of computer code

| Data collection | The Cryo-EM data was collected using software EPU 2. The X-ray crystal structure data was collected in Diamond Light Source (DLS) using DLS developed Generic Data Acquisition (GDA) software (v9.2). The bio-layer interferometry was collected from Octet red 96 using the data requisition software in Octet. The flow cytometry data was acquired from BD Flow cytometer using FACSDIVA (v8.0.1, BD Bioscience). The fluorescence images were acquired with EVOS M5000 1.0.466.664. |
|---|---|
| Data analysis | The software for structure determination and illustration includes Phenix1.18.2-3874, ccp4i2-7.1.012, COOT-0.9.2, UCSF Chimera-1.14, WARP 1.0.6, CryoSPARC-v3.1, UCSF Pyem-v0.5 and Pymol 1.3. Prism 9 was used for statistics and presentation of biochemical and cell based experiments. FlowJo 8.0, NIH Fiji v1.0 and Excel2016 were used for flow cytometry data analysis. |

For manuscripts utilizing custom algorithms or software that are central to the research but not yet described in published literature, software must be made available to editors and reviewers. We strongly encourage code deposition in a community repository (e.g. GitHub). See the Nature Research guidelines for submitting code & software for further information.

## Data

Policy information about availability of data

All manuscripts must include a data availability statement. This statement should provide the following information, where applicable:

- Accession codes, unique identifiers, or web links for publicly available datasets
- A list of figures that have associated raw data
- A description of any restrictions on data availability

The atomic coordinates and structure factors of the crystal structure of DNase I/G-actin/PPP1R15A582-621 complex have been deposited to the PDB with accession code PDB 7NXV. Electron microscope density maps and atomic models of the pre-dephosphorylation complex have been deposited in the EMDB and PDB, respectively, with accession codes EMD-12665 and PDB 7NZM. Structures under PDB accession codes 4MOV, 2A42 and 1KL9 were used as initial models for refinement. Other structures used as alignment for illustration are available in PDB, including 3EKS, 3J8A, 2A19, 6I3M, 6K71, 7D45, 3JAP,1Q46, 6K72, 3CW2, 3JAP.

# Field-specific reporting

Please select the one below that is the best fit for your research. If you are not sure, read the appropriate sections before making your selection.

☒ Life sciences ☐ Behavioural & social sciences ☐ Ecological, evolutionary & environmental sciences

For a reference copy of the document with all sections, see nature.com/documents/nr-reporting-summary-flat.pdf

# Life sciences study design

All studies must disclose on these points even when the disclosure is negative.

| | |
|---|---|
| Sample size | Sample sizes for cell-based experiments are indicated in the relevant figures and the Flow Cytometry section. This number of cells is known to produce a signal-to-noise ratio (in the fluorescence) which is more than sufficient to observe any significant variations between different cell populations. Crystallography and Cryo-EM structures were processed from the best data set after extensive optimization of sample prep and data process. |
| Data exclusions | No data was excluded. |
| Replication | In general, at least 3 biological independent experimental replication were applied, which is commonly accepted to be efficient and practical to show significance and reproducibility. The exact replicate numbers are indicated when relevant. |
| Randomization | Cells were split randomly for different treatment as indicated in FACS experiments. |
| Blinding | No blinding experiments are included. An individual person conducted each experiment according to their expertise. |

# Reporting for specific materials, systems and methods

We require information from authors about some types of materials, experimental systems and methods used in many studies. Here, indicate whether each material, system or method listed is relevant to your study. If you are not sure if a list item applies to your research, read the appropriate section before selecting a response.

### Materials & experimental systems

| n/a | Involved in the study |
|---|---|
| ☒ | ☐ Antibodies |
| ☐ | ☒ Eukaryotic cell lines |
| ☒ | ☐ Palaeontology and archaeology |
| ☒ | ☐ Animals and other organisms |
| ☒ | ☐ Human research participants |
| ☒ | ☐ Clinical data |
| ☒ | ☐ Dual use research of concern |

### Methods

| n/a | Involved in the study |
|---|---|
| ☒ | ☐ ChIP-seq |
| ☐ | ☒ Flow cytometry |
| ☒ | ☐ MRI-based neuroimaging |

## Eukaryotic cell lines

Policy information about cell lines

| | |
|---|---|
| Cell line source(s) | CHO-K1 were sourced from ATCC (CCL-61) and then transduced with a CHOP::GFP reporter as previously reported in PMID:11381086. |
| Authentication | Cells and their derivatives were authenticated by PCR with species specific primers and CRISPR mutagenesis as reported previously (PMID 33220178 and 31749445). |

| Mycoplasma contamination | Negative tested. |
| Commonly misidentified lines (See ICLAC register) | None. |

# Flow Cytometry

## Plots

Confirm that:

☒ The axis labels state the marker and fluorochrome used (e.g. CD4-FITC).

☒ The axis scales are clearly visible. Include numbers along axes only for bottom left plot of group (a 'group' is an analysis of identical markers).

☒ All plots are contour plots with outliers or pseudocolor plots.

☒ A numerical value for number of cells or percentage (with statistics) is provided.

## Methodology

| Sample preparation | Cultured cell line; samples were treated and then collected in PBS-EDTA. |
| Instrument | BD Flowcytometer Fortessas |
| Software | BD Flowcytometer FACSDIVA (v8.0.1, BD Bioscience). |
| Cell population abundance | Approximately 10,000 transfected cells for each sample. |
| Gating strategy | FSC/SSC was used to identify live singlet cells and then 10,000 transfected cells were gated using fluorescent proteins (BFP or mCherry) as indicated in the legends. |

☒ Tick this box to confirm that a figure exemplifying the gating strategy is provided in the Supplementary Information.

