## [Peer Review File · Nature Structural & Molecular Biology]

Peer Review Information

Manuscript Title: Higher order phosphatase-substrate contacts terminate the Integrated Stress Response

Corresponding author name(s): Professor David Ron

Reviewer Comments & Decisions:

Decision Letter, initial version:
--

16th Jun 2021

Dear David,

Thank you again for submitting your manuscript "Higher order phosphatase-substrate contacts terminate the Integrated Stress Response". I apologize for the delay in responding, which resulted from the difficulty in obtaining suitable referee reports. Nevertheless, we now have comments (below) from the 3 reviewers who evaluated your paper. In light of those reports, we remain interested in your study and would like to see your response to the comments of the referees, in the form of a revised manuscript.

You will see that all reviewers were positive about the quality and interest of the work, and reviewer 3 is particularly supportive. However, reviewers 1 and 2 noted a few concerns that should be addressed prior to publication. Reviewer 1 asks to exclude potential confounding effects of variable expression levels of mutant enzymes. The referee also found a discrepancy between cryo-EM map resolutions stated in the manuscript and in the wwPDB validation reports, and further requests to clarify some aspects of the structural model. Moreover, reviewer 1 suggests that the discussion of how the insights obtained extend to other phosphatases should be expanded. Reviewer 2 requests that it should be clarified where the PPP1R15 core segment or full-length PPP1R15 were used. The referee also suggests that more neutral language should be used when discussing the sephin/guanabenz controversy.

Please be sure to respond to all concerns of the referees in full in a point-by-point response and highlight all changes in the revised manuscript text file. If you have comments that are intended for editors only, please include those in a separate cover letter.

We are committed to providing a fair and constructive peer-review process. Do not hesitate to contact

us if there are specific requests from the reviewers that you believe are technically impossible or unlikely to yield a meaningful outcome.

We expect to see your revised manuscript within 6 weeks. If you cannot send it within this time, please contact us to discuss an extension; we would still consider your revision, provided that no similar work has been accepted for publication at NSMB or published elsewhere.

Reporting Summary:

When submitting the revised version of your manuscript, please pay close attention to our [href="https://www.nature.com/nature-research/editorial-policies/image-integrity">Digital Image Integrity Guidelines.](https://www.nature.com/nature-research/editorial-policies/image-integrity)

Data availability: this journal strongly supports public availability of data. All data used in accepted papers should be available via a public data repository, or alternatively, as Supplementary Information. If data can only be shared on request, please explain why in your Data Availability Statement, and also in the correspondence with your editor. Please note that for some data types, deposition in a public repository is mandatory - more information on our data deposition policies and available repositories can be found below:

<https://www.nature.com/nature-research/editorial-policies/reporting-standards#availability-of-data>

[REDACTED]

Kind regards,
Florian

Florian Ullrich, Ph.D.
Associate Editor
Nature Structural & Molecular Biology
ORCID 0000-0002-1153-2040

Referee expertise:

Referee #1: Integrated Stress Response

Referee #2: Kinases/phosphatases

Referee #3: Kinases/phosphatases, structural biology

Reviewers' Comments:

Reviewer #1:

Remarks to the Author:

Yan et al describe structural and biochemical investigation of a key event regulating the integrated stress response in mammalian cells-the recognition of phosphorylated eIF2 alpha by protein phosphatase 1 complex. In the structure led study the authors structure sheds light on the interactions between both and regulatory partners and the substrate that helps to explain why the catalytic PP1c alone is a poor enzyme for the eIF2 substrate as they find that the direct surface interaction with PP1c is minimal and is enhanced by contacts with PPP1R15A. Also present is G Actin that also contacts PPP1R15A. The relevance of the interactions shown in the structure are evidenced by complementary mutagenesis of interface side chains and a combination of in vitro and cellular assays. This presents a nice addition to our knowledge of an important cellular regulatory pathway, however I have several points to raise.

Major

1. In the transfected cell assays shown in Figure 3a how does the expression levels of the various mutant PP1 compare with Wildtype. From the spread of dots up the Y-axis, it appears R588E and probably the double mutants are distributed more towards the lower end of what is a log plot. As lower enzyme levels might correlate with lower activity, it would appear appropriate to consider the impact of expression only within narrower window of comparable levels of each construct to be sure that the assay is not being affected by differing expression levels.
2. Similarly does the expression of plasmid copy of PPP1R15A-mCherry have any impact on the endogenous PPP1R15A or B levels that could contribute to the result shown?
3. The CryoEM validation report for 7NZM FSC plot estimates the resolution as 4.87-4.88 (Table 8.2), while the equivalent plot in Fig S2 suggests the resolution reported in the main text. Can the authors reconcile this?
4. In Figure S2 e where the eIF2alpha phosphorylation loop is modelled into the density mesh, what happened to the side chains for L50 and R52? If no clear density, this should be indicated.
5. Figure 7 makes some very good comparisons with the PKR-eIF2 structure and extends the mutagenesis to Perk. These experiments show the role of a conserved surface patch on eIF2alpha identified originally in PKR that here is also important for PP1 binding. There are a number of other eIF2-alpha structures and co-structures, some of which are referred to briefly at the bottom of page 11 by PDB reference numbers, (not article citations[which would be appropriate]). It would be helpful to expand this discussion/comparison.
 - (i) It appears from the text/images that the phospho-loop adopts a distinct conformation in this structure. A figure overlaying a diverse sample of these would be informative.
 - (ii) Perhaps as interesting as the comparison with PKR, recent eIF2-eIF2B co-structures (along with historical mutagenesis in the yeast system by Dever and colleagues) showed that the conserved K(79)GYID(83) sequence mutated here also forms the interface with eIF2B. This indicates that all the major players involved in writing, reading or erasing the Ser51P modification are all docking via this same remote site.
 - (iii) As there are now several structures of the eIF2 homo-trimer (including bound to initiator tRNA and 40S ribosomes) it should be possible to overlay/dock the other eIF2 subunits to this new structure to determine if any of the other parts of eIF2 that are found in vivo clash with any of the elements in the new structure.

6. As phosphorylated eIF2 binds tightly to eIF2B in cells via the same motif identified here (see 5.(ii) above), how do the authors propose that PP1C can gain access to eIF2alphaP to mediate control in cells?

minor points

1. p6 first paragraph, it was not clear if the actin shown in Figure 1e is from the new structure or from the pdb file 4V0U as labelled as 4V0U is claimed to be very poor resolution.
2. In figures (Figs 2, 6 and 7) where baseline phosphatase activity is measured it would be helpful to show these on a separate plot as the scale used does not facilitate cross-comparison
3. In the transfected cell assays shown in Figure 3a and 4a what is shown in red in these plots? the legend is not informative.
4. It is hard to reconcile the spread of x-axis CHOP::GFP signals in Figure 3a/4a (at least an order of magnitude (shown on a log scale) with the tight error bars shown in Figure 3b/4b (on a linear scale). What error is being shown?

Reviewer #2:

Remarks to the Author:

In the manuscript, Higher order phosphatase-substrate contacts terminate the Integrated Stress Response, authors Yan et al present structures of a core segment of PPP1R15A/B lacking the disordered N terminus together with Actin and then a cryoEM structure of the tripartite complex: PP1R15 (core) + actin + EIF2alpha (the substrate). Both structures also include DNaseI bound to G-actin.

This work provides novel insights into how PPP1R15 and Actin work together to coordinate interactions with the substrates, and show how the regulated phosphosite (pS51) is targeted to the catalytic site for dephosphorylation. Two particularly interesting and significant implications are discussed--first how actin surfaces that interact w/ PPP1R15 are conserved even in organisms that lack PPP1R15, i.e. the phosphatase evolved to utilize preexisting surfaces, and second how the pre-phosphorylation complex and pre-dephosphorylation complexes differ and are incompatible, due to a key trp residue being either in 'trp up' or 'trp down' positions. This has implications for termination of the phosphoreaction and 'handoff' to the phosphatase.

This paper is significant interesting well written. The data is rigorous and well presented. Analyses are thorough in combining both in vivo and in vitro approaches and key aspects of the model are thoroughly tested through mutagenesis of complementary components and surfaces, including key mutation in actin. To complement structural analyses, binding assays using biolayer interferometry are also used.

I have only 2 significant suggestions/comments for improvement:

--Authors should be clear in the manuscript text when a core segment of PPP1R15 is used (i.e. in the structures and in vitro dephosphorylation assays) vs when full length R15 including the N terminus is used (all in vivo assays).

--Second, the controversy about mechanism of Sephin and guanabenz is well known in the phosphatase field. The authors make a somewhat snide statement about this in the discussion: "Mechanism-based pharmacological targeting of substrate-specific holophosphatases is a long-sought

goal, which, despite claims to contrary⁴³, may have yet to be realised^{26,27}.". This is not useful for readers, especially if you hope to attract a broad readership. I suggest more neutral language acknowledging that there is some controversy in this area. Obviously the structures presented here could lead to development of different small molecule inhibitors of the complex which would be a significant addition to the field. No need to denigrate other colleagues and their findings in print.

Reviewer #3:

Remarks to the Author:

Review of Yan and coworker's manuscript entitled 'Higher order phosphatase-substrate contacts terminate the Integrated Stress Response'

The authors report complementary X-ray crystal and cryo-EM structures that provide a detailed snapshot of the PP1A/PPP1R15A/G-actin holo-enzyme in action dephosphorylating the translation initiation factor eIF2 α on its Ser51 phospho-regulatory site.

Described are a 2.55 Angstrom X-ray crystal structure of a PPP1R15A/G-actin/DNase I complex and a 3.96 Angstrom cryo-EM structure of a PP1A/PPP1R15A/G-actin/phosho-eIF2 α pre-dephosphorylation complex.

Notable tricks that allowed for successful structure determinations include:

1. Exploiting the Asp64Ala mutation to trap an otherwise transient complex between enzyme and substrate.
2. Use of DNase 1 to increase the stability of X-ray and cryo-EM samples.

The authors perform compelling in vitro enzyme and binding studies and in cellulo functional studies (flow cytometry and stress granule assays) to validate what they see in their atomic structure analyses are functionally relevant.

One particularly fun observation was how the authors new structures accounted for non-modelled residual density in previously solved structures. This and how all the pieces of the structural puzzle fit together in such a sensical manner gave confidence that the author's enzyme-substrate recognition model is correct.

The manuscript is well written and easy to follow. The presented experimental data are of high quality and well controlled, the interpretations appear sound, and the proposed model for eIF2 α substrate recognition is compelling. I recommend publication of the presented work with great enthusiasm.

Overall, I have no significant criticisms of the presented work but as I am not an expert cryo-EM spectroscopist, I defer to other reviewers to provide feedback on that specific aspect of the storyline.

Author Rebuttal to Initial comments

9-Jul-2021 (V1.4)

Point-by-point response to reviews of NSMB-A44897A-Z

Reviewer #1:

We thank the reviewer for their detailed analysis of the work and for their insightful comments

Major

1. In the transfected cell assays shown in Figure 3a how does the expression levels of the various mutant PP1 compare with Wildtype. From the spread of dots up the Y-axis, it appears R588E and probably the double mutants are distributed more towards the lower end of what is a log plot. As lower enzyme levels might correlate with lower activity, it would appear appropriate to consider the impact of expression only within narrower window of comparable levels of each construct to be sure that the assay is not being affected by differing expression levels.

We agree.

In the revised manuscript, we have confined the analysis of reporter gene activity to cells expressing comparable levels of the different PPP1R15A effector proteins. Defective antagonism of the ISR remains a conspicuous feature of the double mutant F585A;R588E and F585A;R588A derivative even after eliminating the brightest PPP1R15A-mCherry expressing cells - which were a feature of some, but not all PPP1R15A derivatives (as noted by the reviewer).

2. Similarly does the expression of plasmid copy of PPP1R15A-mCherry have any impact on the endogenous PPP1R15A or B levels that could contribute to the result shown?

Deregulated expression of an active PPP1R15A regulatory subunit results in strong attenuation of the endogenous gene see figure 1A of Novoa et al., 2003¹, this is expected, given the role of PPP1R15 in negative feedback in the ISR. This feature would tend to minimise the difference in ISR attenuation wrought by PPP1R15A derivatives and thus introduces an experimental bias against the hypothesis being tested, strengthening the conclusion that the various PPP1R15A derivatives tested are indeed intrinsically different in their ability to attenuate the ISR. We thank the reviewer for raising this issue.

3. The CryoEM validation report for 7N2M FSC plot estimates the resolution as 4.87-4.88 (Table 8.2), while the equivalent plot in Fig S2 suggests the resolution reported in the main text. Can the authors reconcile this?

These differences arise from the different resolution calculated in presence or absence of the mask. In revised supplementary Fig. 2a we have introduced arrows into the FSC curve to highlight this point. We have also deposited the mask to the EMDDB resource (section 6.5 of the validation report).

4. In Figure S2 e where the eIF2alpha phosphorylation loop is modelled into the density mesh, what happened to the side chains for L50 and R52? If no clear density, this should be indicated.

The figure legend indicates that eIF2 Arg52 sidechain was not modelled as there is no corresponding density and eIF2 α Leu50 sidechain is not shown for clarity

5. Figure 7 makes some very good comparisons with the PKR-eIF2 structure and extends the mutagenesis to Perk. These experiments show the role of a conserved surface patch on eIF2alpha identified originally in PKR that here is also important for PP1 binding. There are a number of other eIF2-alpha structures and co-structures, some of which are referred to briefly at the bottom of page 11 by PDB reference numbers, (not article citations[which would be appropriate]). It would be helpful to expand this discussion/comparison.

We have also included references to the specific papers next to the PDB references on page 11

(i) It appears from the text/images that the phospho-loop adopts a distinct conformation in this structure. A figure overlaying a diverse sample of these would be informative.

In response to this comment, we have now included an additional panel, **f** in supplementary Fig. S2. There we have aligned the S51 loop from diverse structures. In the assortment of available structures, the flipped-out orientation of Ser51p stands out as a feature unique to the predephosphorylation complex.

(ii) Perhaps as interesting as the comparison with PKR, recent eIF2-eIF2B co-structures (along with historical mutagenesis in the yeast system by Dever and colleagues) showed that the conserved K(79)GYID(83) sequence mutated here also forms the interface with eIF2B. This indicates that all the major players involved in writing, reading or erasing the Ser51P modification are all docking via this same remote site.

In response to these insightful comments, in the revised *Discussion* we have now addressed the exploitation of residues on eIF2 $\mathcal{L}_{4,5}$ and \mathcal{I}_5 in complex formation with eIF2B and added the speculation that whilst eIF2B has a high affinity to eIF2-P Kd as low as 3 nM², the high turnover of the complex³, likely favours the one-way flow of substrate from eIF2B to the phosphatase.

(iii) As there are now several structures of the eIF2 homo-trimer (including bound to initiator tRNA and 40S ribosomes) it should be possible to overlay/dock the other eIF2 subunits to this new structure to determine if any of the other parts of eIF2 that are found in vivo clash with any of the elements in the new structure.

New supplementary Fig. 3 shows that an entire eIF2 trimer can be docked into the predephosphorylation complex with no hindrance imposed by the β and β' subunits (absent from the complex assembled here)

6. As phosphorylated eIF2 binds tightly to eIF2B in cells via the same motif identified here (see 5.(ii) above), how do the authors propose that PP1C can gain access to eIF2 α P to mediate control in cells?

Please see response above to point 5ii

minor points

1. p6 first paragraph, it was not clear if the actin shown in Figure 1e is from the new structure or from the pdb file 4V0U as labelled as 4V0U is claimed to be very poor resolution.

Both the PP1-R15B and the R15A-Actin shown in ribbon diagram are from high resolution structures (PDB 4V0X and the crystal structure here, PDB 7NXV). The low resolution ternary PP1/R15B/Actin structure (PDB 4V0U) is the scaffold into which the two high resolution structures have been aligned. We have modified the legend to clarify this point.

2. In figures (Figs 2, 6 and 7) where baseline phosphatase activity is measured it would be helpful to show these on a separate plot as the scale used does not facilitate cross-comparison

In response to this helpful comment we have presented the data in figure 2b, 6c and 7b-lower panel with a discontinuous 'Y' axis.

3. In the transfected cell assays shown in Figure 3a and 4a what is shown in red in these plots? the legend is not informative.

In addition to the plasmid encoding the PPP1R15A::mCherry effector, the cells in figure 3 were also transfected with a Blue Fluorescent Protein (BFP) expressing 'empty' version of the same plasmid that is later used to deliver the beta-Actin gene (in figure 4). This was done to favour comparability with the FACS experiments shown in both figures. Thus, BFP expression serves as an additional fiduciary of transfection. In the original version of the figure, the plot of the BFP positive (Blue) cells were overlaid with the plot of BFP negative (red) cells that represented a population that has failed to take up the plasmid DNA in the transfection mix. However, as the BFP negative population is not analysed for its PPP1R15A-mCherry or CHOP::GFP signals, we have eliminated this unnecessary overlay and have provided the gating strategy in new supplementary Fig. 4, which demonstrates the typical ratio of BFP+ to BFP- cells. In addition, and as per NSMB policy, we have updated the flow cytometry figures to use contour plots (with outliers) to show the abundance of cells with different mCherry-fusion protein expression levels. We have amended the figure legend accordingly and thank the reviewer for drawing our attention to this point

4. It is hard to reconcile the spread of x-axis CHOP::GFP signals in Figure 3a/4a (at least an order of magnitude (shown on a log scale) with the tight error bars shown in Figure 3b/4b (on a linear scale). What error is being shown?

The large number of data points acquired in each FACS experiment renders the error in measurement of the mean (SEM) very small. Thus, confidence in the numeric value of median of the reporter activity in any given FACS measurement, which constitute a single data point in the analysis shown in panel 'c', is high.

In light of the reviewer's comment, we have revised the presentation of the data in 3c, 4c and 6e to a 'box and whiskers' plot. Like the bar diagram in the original version, the box and whiskers plot reports on the median value of the population (the grey bar across the box), but also on the 25th, 75th and 99th percentile values (lower and upper limits of the box and ends of the whiskers).

Reviewer #2:

We thank the reviewer for their assessment of our work and for their suggestions for improvements

--Authors should be clear in the manuscript text when a core segment of PPP1R15 is used (i.e. in the structures and in vitro dephosphorylation assays) vs when full length R15 including the N terminus is used (all in vivo assays).

We thank the reviewer for alerting us to this ambiguity. In the revised manuscript we now emphasize that the experiments conducted in cells make use of the full-length regulatory subunit.

--Second, the controversy about mechanism of Sephin and guanabenz is well known in the phosphatase field. The authors make a somewhat snide statement about this in the discussion: "Mechanism-based pharmacological targeting of substrate-specific holophosphatases is a long-sought goal, which, despite claims to contrary⁴³, may have yet to be realised^{26,27}". This is not useful for readers, especially if you hope to attract a broad readership. I suggest more neutral language acknowledging that there is some controversy in this area. Obviously the structures presented here could lead to development of different small molecule inhibitors of the complex which would be a significant addition to the field. No need to denigrate other colleagues and their findings in print.

We have no intention of making snide comments or denigrating colleagues. We do however feel that our perspective on the aforementioned controversy is relevant to the significance of the findings presented here and we wish to incorporate that perspective succinctly, politely but emphatically in this paper. With that in mind, we have modified the text. The previous version stated – in so many words and as a matter of fact – that the compounds believed by some to be specific inhibitors of the PPP1R15A-containing holophosphatase lack such activity. In the revised version we have introduced the

important nuance that this is *our* opinion. We believe this edit now aligns the text closely with the facts – for it is certainly a fact that whilst other may hold different opinions on the matter *we* believe these compounds lack significant activity as holophosphatase inhibitors.

References

1. Novoa, I., Zhang, Y., Zeng, H., Jungreis, R., Harding, H.P. & Ron, D. Stress-induced gene expression requires programmed recovery from translational repression. *Embo J* **22**, 1180-7 (2003).
2. Jennings, M.D., Kershaw, C.J., Adomavicius, T. & Pavitt, G.D. Fail-safe control of translation initiation by dissociation of eIF2alpha phosphorylated ternary complexes. *Elife* **6**(2017).
3. Zyryanova, A.F., Kashiwagi, K., Rato, C., Harding, H.P., Crespillo-Casado, A., Perera, L.A., Sakamoto, A., Nishimoto, M., Yonemochi, M., Shirouzu, M., Ito, T. & Ron, D. ISRIB Blunts the Integrated Stress Response by Allosterically Antagonising the Inhibitory Effect of Phosphorylated eIF2 on eIF2B. *Mol Cell* **81**, 88-103 e6 (2021).

Decision Letter, first revision:

22nd Jul 2021

Dear David,

Thank you again for submitting your revised manuscript "Higher order phosphatase-substrate contacts terminate the Integrated Stress Response" (NSMB-A44897B). It has now been seen by two of the original referees and their comments are below. The reviewers find that the paper has improved in revision, and therefore we'll be happy in principle to publish it in Nature Structural & Molecular Biology, pending minor revisions to satisfy the referees' final requests and to comply with our editorial and formatting guidelines.

We are now performing detailed checks on your paper and will send you a checklist detailing our editorial and formatting requirements in about a week. Please do not upload the final materials and make any revisions until you receive this additional information from us. To facilitate our work at this stage, we would appreciate if you could send us the main text as a word file. Please make sure to copy the NSMB account (cc'ed above).

Kind regards,
Florian

Florian Ullrich, Ph.D.
Associate Editor
Nature Structural & Molecular Biology
ORCID 0000-0002-1153-2040

Reviewer #1 (Remarks to the Author):

The authors have addressed all the substantive points made in the original review.

In the new text (page 17) and the rebuttal (point 5ii) the authors make use of a cursive L to refer to a part of the structure. I have not seen that symbol used before. Perhaps edit to use the relevant residue number range?

Reviewer #2 (Remarks to the Author):

I have reviewed the revised submission from David Ron and colleagues and feel they have adequately addressed all the reviewers' comments.
Martha Cyert

Final Decision Letter: